# E²LoRA: Efficient and Effective Low-Rank Adaptation with Entropy-Guided Adaptive Sharing

**Minglei Li**[1] *  **Peng Ye**[1,2,3] *  **Jingqi Ye**[2,4]  **Haonan He**[2,4]  **Tao Chen**[1,5] †

[1] Fudan University    [2] Shanghai Artificial Intelligence Laboratory
[3] The Chinese University of Hong Kong    [4] University of Science and Technology of China
[5] Shanghai Innovation Institute

## Abstract

As large pre-trained models rapidly scale, Parameter-Efficient Fine-Tuning (PEFT) through methods like Low-Rank Adaptation (LoRA) becomes increasingly crucial. While LoRA has emerged as a cornerstone of PEFT, excelling at preserving performance with minimal additional parameters, exploring parameter-sharing mechanisms of LoRA remains critical to pushing efficiency boundaries. However, existing naive LoRA sharing methods often degrade performance due to sacrificed representational diversity and weakened model expressiveness. To overcome this issue, we conduct an in-depth analysis of pre-trained models using gradient-based proxy entropy, and uncover two critical, previously overlooked properties: Local Similarity and Layer-wise Information Heterogeneity. Building on these insights, we propose E²LoRA, a novel dual-adaptive sharing framework. It enables adaptive sharing interval partitioning, guided by inter-layer proxy entropy similarity, and adaptive rank allocation, informed by layer-wise absolute proxy entropy. This unique design leverages inherently informative properties of pre-trained models to significantly reduce parameter redundancy while maintaining or enhancing expressiveness. Comprehensive evaluations across diverse tasks, modalities, and models consistently demonstrate that E²LoRA achieves an excellent balance of efficiency and effectiveness, consistently matching or surpassing baselines with approximately 50% fewer trainable parameters.

## 1 Introduction

As large language models (LLMs) scale, *full fine-tuning* becomes increasingly resource-intensive. *Parameter-Efficient Fine-Tuning* (PEFT) methods emerge as alternative techniques, efficiently adapting models by updating only a small set of parameters, significantly reducing training costs (Zaken et al., 2021; Liu et al., 2021; Hu et al., 2021). Among PEFT methods, *Low-Rank Adaptation* (LoRA) (Hu et al., 2021) stands out. By injecting pluggable low-rank adapters, its simplicity and effectiveness have driven its broad adoption across industries and research. As a result, numerous follow-up studies and variants are proposed (Mao et al., 2025; Han et al., 2024).

Currently, LoRA variants can be roughly classified into two main categories: (1) *Effectiveness-enhancing variants* such as AdaLoRA (Zhang et al., 2023c) dynamically adjust ranks, DoRA (Liu et al., 2024a) restructures LoRA through decomposition, and LoRA+ (Hayou et al., 2024) optimizes learning dynamics via asymmetrical learning rates. However, these methods focus on performance gains without enhancing parameter efficiency; (2) *Efficiency-enhancing methods* such as LoRA-FA (Zhang et al., 2023b), which freezes the $A$ matrices and keeps the $B$ matrices trainable, VeRA (Kopiczko et al., 2023) adds learnable vectors at the middle and output ends of LoRA, freezes and shares the low-rank matrices, and only updates the introduced vectors, Tied-LoRA (Renduch-intala et al., 2023) further investigates which parts of LoRA should be shared or frozen. Besides,

---

*Equal contribution.
†Corresponding author.

ShareLoRA (Song et al., 2024) shares $A$ matrix across all layers while keeping $B$ matrices layer-specific. While these methods reduce parameters through freezing or sharing, they often incur significant performance degradation. This is primarily because sharing the same LoRA parameters across all layers leads to a loss of representational diversity, disrupting feature discrimination and weakening the model's expressiveness. This raises a critical question: *How can we enhance the efficiency of LoRA significantly while maintaining or even surpassing original effectiveness?*

To answer this question, we draw inspiration from information theory and conduct an in-depth analysis of pre-trained models using gradient-based proxy entropy. As depicted in Figure 1, our analysis reveals two crucial insights often overlooked by prior sharing strategies: (1) **Local Similarity:** The left "Entropy Similarity Matrix" panels of Figure 1 show the inter-layer information similarity of gradients on different models and tasks. We observe distinct block-like high-similarity regions, which are highlighted by green boxes. This phenomenon indicates that adjacent layers often exhibit higher information similarity. Moreover, the size and position of these similarity blocks vary significantly across different models and tasks, implying that heuristic sharing ranges like sharing across all layers or fixed blocks are suboptimal. (2) **Layer-wise Information Heterogeneity:** The right "Per-Layer Entropy Comparison" panels of Figure 1 illustrate the absolute proxy entropy values of different layers. We observe substantial variations in the volume of information across layers. This heterogeneity means that after defining sharing intervals, the informational importance

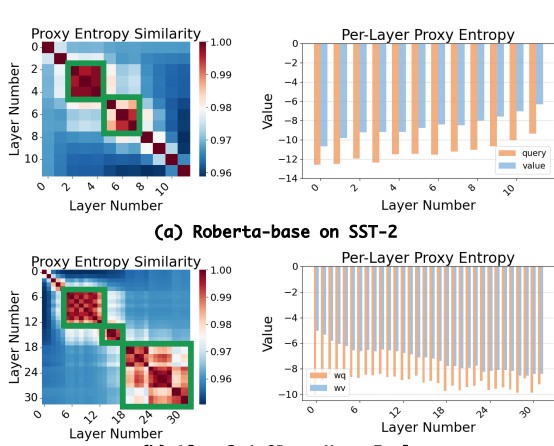

(a) Roberta-base on SST-2

(b) Llama3.1-8B on HumanEval

Figure 1: Gradient-based proxy entropy analysis for pre-trained models. Left panels show inter-layer information similarity, with high-similarity blocks highlighted in green. Right panels illustrate absolute proxy entropy values across layers, indicating significant variations. Subfigures (a), (b) present the results of Roberta-base on SST-2 and Llama3.1-8B on HumanEval, respectively. Visualization of CLIP-ViT-B/16 on Cars can be found in the Appendix D.1.

differs significantly across intervals. Applying a uniform number of learnable parameters to intervals with unequal information content may hinder the performance of information-rich layers, as they are not allocated sufficient capacity. Prior methods overlook these architectural heterogeneities, leading to information capacity loss and performance degradation when parameters are shared indiscriminately. Critically, these structural properties are not unique to the models in Figure 1. As shown in Appendix D.2, we confirm the universality of these insights across a broader range of models, including Qwen3-8B, Phi4-mini, Gemma-9B-it, and GPT-OSS-20B, which cover different model families and architectures.

Building on these critical insights, we introduce $E^2$LoRA, a novel dual-adaptive parameter-sharing framework that effectively addresses the aforementioned challenges. For *Local Similarity*, $E^2$LoRA dynamically partitions layers into sharing intervals by measuring inter-layer similarity with a gradient-based entropy proxy and applying an adaptive threshold. This strategy groups consecutive layers only when they exhibit strong informational resemblance, avoiding heuristic schemes and thereby improving parameter efficiency without sacrificing model expressiveness. For *Layer-wise Information Heterogeneity*, $E^2$LoRA adopts an entropy-driven proxy strategy for rank allocation. We utilize the absolute gradient proxy entropy of each layer as a direct measure of its informational significance, guiding adaptive rank assignment across intervals. Layers with richer information are allocated higher ranks to preserve expressiveness, while less critical layers use lower ranks. Overall, $E^2$LoRA achieves an optimized trade-off between efficiency and effectiveness.

Our contributions can be summarized as follows:

- **Novel Insights into Parameter Sharing:** Using a gradient-based proxy entropy analysis, we reveal two previously overlooked properties of pre-trained models: (1) *Local Similarity*—contiguous regions where adjacent layers exhibit high information overlap, and (2) *Layer-wise Information Heterogeneity*—substantial variation in information content across layers.

These insights highlight the limitations of existing parameter-sharing methods that rely on uniform or heuristic patterns while ignoring such structural heterogeneities.

- **Dual Adaptive Parameter Sharing Framework:** Motivated by these insights, we propose $E^2$LoRA, a novel framework that jointly optimizes adaptive parameter sharing and LoRA rank allocation. To address *Local Similarity*, $E^2$LoRA partitions layers into adaptive sharing intervals based on gradient information similarity. To tackle *Layer-wise Information Heterogeneity*, it leverages absolute gradient proxy entropy to dynamically assign LoRA ranks per sharing interval. This dual-adaptive design enables highly efficient parameter utilization while preserving or even enhancing model performance.

- **Proven Efficiency-Effectiveness Balance:** We conduct extensive evaluations for the proposed $E^2$LoRA across diverse tasks, models, and modalities. Experimental results demonstrate that $E^2$LoRA strikes an excellent balance of parameter efficiency and performance, consistently matching or surpassing sharing and non-sharing baselines while significantly reducing the number of trainable parameters by approximately 50%.

## 2 RELATED WORK

**Parameter-sharing LoRA Variants** have emerged as a promising direction for enhancing the efficiency and effectiveness of LoRA. Various approaches have been proposed to optimize different aspects: ShareLoRA (Song et al., 2024) introduces cross-layer sharing of low-rank matrices (A, B, or both). VeRA (Kopiczko et al., 2023) implements a more constrained approach by sharing fixed low-rank matrices across layers, updating only layer-specific scaling vectors. Beyond efficiency, some sharing strategies also target performance improvements. Rasa (He et al., 2025b) shares a certain ratio of ranks from each adapter, concatenating shared and unique low-rank weights to increase overall capacity, while RandLoRA (Albert et al., 2025) shares full-rank random bases across layers and introduces trainable diagonal scaling matrices to enable high-rank fine-tuning. HydraLoRA (Tian et al., 2024) further adopts an asymmetric shared-$A$ design within a router-based architecture to mitigate task interference on heterogeneous data, and BSLoRA (Zhou et al.) decomposes parameters into local, intra-layer, and globally shared components to build a fixed hierarchical sharing structure. In contrast to these methods with manually specified sharing topologies, our approach derives sharing intervals directly from task gradients via entropy and similarity statistics.

**Rank-Allocation LoRA Variants** enhance the expressiveness of LoRA by adaptively allocating ranks. For instance, AdaLoRA (Zhang et al., 2023c) begins training with an over-parameterized rank budget and progressively prunes less important ranks. Conversely, IncreLoRA (Zhang et al., 2023a) starts with rank-1 adapters and gradually allocates higher ranks to adapters with higher importance, avoiding the extra cost overhead of over-parameterization. ALoRA (Liu et al., 2024b) and DyLoRA (Valipour et al., 2023) also adjust ranks dynamically, either pruning and reallocating capacity or supporting search-free flexible ranks at inference time, while RA-LoRA (Kim et al., 2024) focuses on quantized LLMs by assigning higher ranks to layers with larger quantization errors. LoRA-GA (Wang et al., 2024a) and LoRA-One (Zhang et al., 2025) further exploit gradient information to guide rank or parameter allocation, bringing LoRA closer to full fine-tuning behavior. GoRA (He et al., 2025a) sets layer-wise ranks upfront via gradient importance from pre-trained weights. This avoids dynamic shape adjustment during training, benefiting distributed strategies like ZeRO (Rajbhandari et al., 2020). RaLoRA (Ye et al., 2026) adaptively aligns ranks with the intrinsic dimensionality of layer-wise gradients. All of these methods primarily adjust ranks within layers, whereas our $E^2$LoRA combines inter-layer parameter sharing with entropy-guided rank allocation to jointly optimize efficiency and effectiveness.

**Entropy and its variants** have been widely used in deep learning as measures of information content. Prior work (Lin et al., 2024; Yang et al., 2025; Lu et al., 2024) has employed inter-layer entropy to guide model pruning, simplifying architectures without performance degradation. In addition, entropy-based similarity has been leveraged in tasks such as knowledge distillation (Jiao et al., 2019), model comparison (Hoyos-Osorio & Sanchez-Giraldo, 2023; Ali et al., 2025), and attention optimization (Zhai et al., 2023), demonstrating strong empirical effectiveness. Inspired by these successes, and building upon the computational forms for entropy established in related literature, our work is the first to explore the use of such entropy-like measures and their derived similarities in the context of LoRA. Specifically, we utilize these proxy entropy-based features, ex-

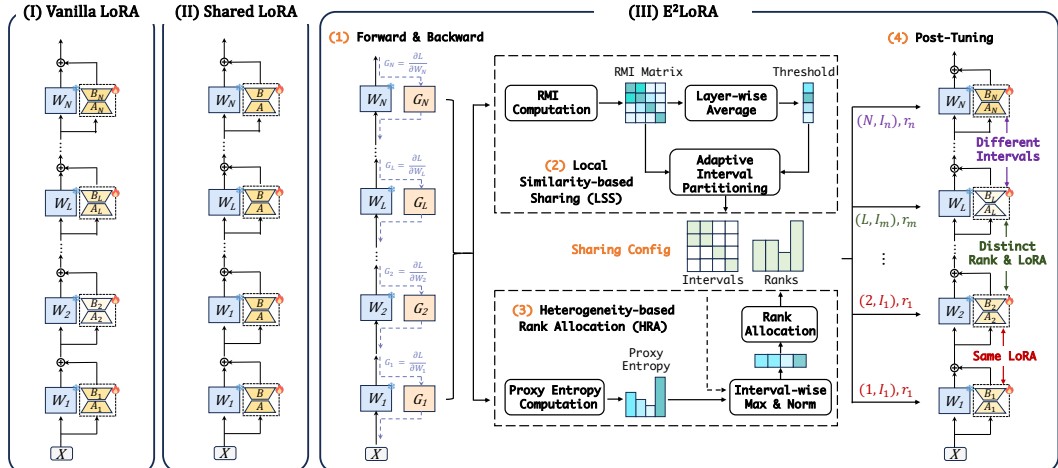

Figure 2: Overall E²LoRA framework. (I) shows Vanilla LoRA. (II) depicts LoRA shared globally. (III) details E²LoRA: (1) Layer gradients are first obtained. (2) Local Similarity-based Sharing utilizes RMI computation and adaptive interval partitioning to configure sharing groups. (3) Heterogeneity-based Rank Allocation employs entropy computation to assign appropriate ranks. (4) Finally, these adaptively configured LoRA adapters are applied during post-tuning.

tracted from downstream task gradient data, to adaptively guide parameter sharing—a previously unexplored direction for LoRA.

## 3 METHOD

Our proposed framework E²LoRA successfully optimizes the trade-off between parameter efficiency and model performance in LoRA-based approaches. As shown in Figure 2, the core of our method includes two components: *Local Similarity-based Sharing (LSS)* and *Heterogeneity-based Rank Allocation (HRA)*. The overall flow is detailed in Algorithm 1. We will introduce these two parts in detail below, and then introduce how to combine E²LoRA with different LoRA variants.

### 3.1 LOCAL SIMILARITY-BASED SHARING (LSS)

To facilitate effective parameter sharing while preserving representational diversity, we first identify groups of layers in the pre-trained model that exhibit high similarity in their information content before fine-tuning. This process consists of two key steps: (1) quantifying inter-layer similarities by analyzing information of the gradient, and (2) adaptively partitioning layers into sharing intervals based on these similarity measures.

**Quantifying Inter-Layer Gradient Similarity.** We hypothesize that gradient information, derived from specific downstream tasks, reflects the similarity of information across different layers and can guide more efficient parameter sharing. Following (Lin et al., 2024), we define an entropy-like metric to quantify the information spread within the gradients of each layer. For a given layer $l$, let $\mathbf{G}_l$ denote its gradients obtained on a mini-subset of the training dataset. We calculate this measure, which we term 'proxy entropy' $H(\mathbf{G}_l)$, as:

$$H(\mathbf{G}_l) = \log(\sigma_{\mathbf{G}_l}) + \frac{1}{2}\log(2\pi) + \frac{1}{2}, \tag{1}$$

where $\sigma_{\mathbf{G}_l}$ is the standard deviation of the elements in the flattened gradient tensor $\mathbf{G}_l$. This approximation effectively captures the spread and uncertainty with the layer's gradients. We provide a detailed theoretical justification connecting this proxy entropy to the Frobenius norm and Fisher Information in Appendix B.

To assess the redundancy or similarity in information between two layers $i$ and $j$, we compute the Mutual Information (MI) between their respective gradient tensors, $\mathbf{G}_i$ and $\mathbf{G}_j$, based on the proposed proxy entropy. Mutual Information measures the amount of information shared between

two variables, and it is defined as:

$$I(\mathbf{G}_i; \mathbf{G}_j) = H(\mathbf{G}_i) + H(\mathbf{G}_j) - H(\mathbf{G}_i, \mathbf{G}_j). \tag{2}$$

Here, $H(\mathbf{G}_i, \mathbf{G}_j)$ represents the joint entropy of the gradient tensors $\mathbf{G}_i$ and $\mathbf{G}_j$. To compute this, we flatten $\mathbf{G}_i$ and $\mathbf{G}_j$, concatenate them, and then approximate $H(\mathbf{G}_i, \mathbf{G}_j)$ using equation 1. A higher mutual information value indicates greater shared information and potential redundancy between the task-specific functions of two layers.

Since MI is an absolute quantity, it does not reflect the proportion of shared information. To provide a normalized and comparable measure of similarity of different layer pairs, we adopt Relative Mutual Information (RMI):

$$RMI(\mathbf{G}_i; \mathbf{G}_j) = \frac{I(\mathbf{G}_i; \mathbf{G}_j)}{\min(H(\mathbf{G}_i), H(\mathbf{G}_j))}. \tag{3}$$

RMI provides a normalized measure of similarity, ranging from 0 to 1, making it suitable for comparing similarities across different layer pairs regardless of their absolute information content.

**Adaptive Sharing Interval Partitioning.** With the inter-layer RMI values, we proceed to dynamically partition the pre-trained model's layers into variable-length sharing intervals. This adaptive partitioning respects the "Local Similarity" property identified in the Introduction, where adjacent layers often exhibit higher similarity, but the range of this similarity varies and is not fixed. Our goal is to group layers that are informationally similar into contiguous blocks, allowing parameter sharing within these blocks.

Our partitioning strategy is a greedy, sequential process guided by self-adaptive, layer-specific similarity thresholds. For each layer $\mathbf{G}_m$ in a model with $N$ layers, we first compute the Relative Mutual Information (RMI) between $\mathbf{G}_m$ and every other layer $\mathbf{G}_k$ (where $k \neq m$). A unique threshold $\tau_m$ for layer $\mathbf{G}_m$ is then determined by averaging these computed RMIs:

$$\tau_m = \frac{1}{N-1} \sum_{k=1, k \neq m}^{N} RMI(\mathbf{G}_m, \mathbf{G}_k). \tag{4}$$

Here, the layer-specific $\tau_m$ serves as a dynamic baseline to identify layers sufficiently similar to $\mathbf{G}_m$, guiding the adaptive partitioning of consecutive layers into sharing intervals. The partitioning algorithm then proceeds greedily and sequentially. Starting a new interval from the current layer, it expands by including subsequent layers as long as their RMI with the interval's starting layer meets the determined threshold. Once the similarity condition is no longer met, the current interval is finalized, and a new one commences. This iterative process continues until all layers are assigned to a non-overlapping sharing interval, as detailed in Algorithm 2.

This process results in a set of non-overlapping, consecutive sharing intervals, each containing layers that exhibit a high information similarity via their gradient properties. We further validate this design in Appendix G by comparing it against a globally optimal Dynamic Programming (DP) solution, demonstrating that our greedy strategy achieves comparable performance while being significantly faster and eliminating the need for hyperparameter tuning.

## 3.2 HETEROGENEITY-BASED RANK ALLOCATION (HRA)

The second crucial component of $\text{E}^2\text{LoRA}$ is the adaptive allocation of LoRA ranks, addressing the Layer-wise Information Heterogeneity property identified in the Introduction. While the previous section detailed how to group layers into sharing intervals to reduce parameters, this section focuses on distributing the available LoRA rank capacity among these intervals intelligently. Our goal is to assign more expressive power, i.e., higher ranks, to intervals that carry more critical information, thus preserving overall performance while maintaining parameter efficiency.

For each sharing interval $[s_k, e_k]$ identified by the Adaptive Sharing Interval Partitioning algorithm, we need to determine a representative importance score. Since each interval will be associated with a single shared LoRA adapter, this adapter's rank should reflect the information content of the layers it represents. We utilize the gradient proxy entropy $H(\mathbf{G}_l)$ calculated for each layer defined in equation 1 as a measure of its information content. To ensure that even the most information-rich layer within an interval, we define the representative proxy entropy for an interval $k$ as the maximum

---

**Algorithm 1:** Overall Algorithm for E²LoRA Framework

---

**Input:** Model gradients $\{\mathbf{G}^{(1)}, \ldots, \mathbf{G}^{(N)}\}$,
Number of layers $N$, Base rank $r_{vanilla}$
**Output:** Sharing intervals $\mathcal{S} = \{[s_k, e_k]\}$, Allocated ranks $\mathcal{R} = \{r_k\}$
▷ Step 1: Compute Local Similarity-based Sharing (LSS)
**for** *each layer l* **do**
    Compute proxy entropy $H(\mathbf{G}_l)$ ;                   `// based on equation 1`
    Compute pairwise $\mathrm{RMI}(\mathbf{G}_i, \mathbf{G}_j)$ ;          `// based on equation 3`
▷ Step 2: Adaptive Partitioning
**for** *each layer l* **do**
    Compute threshold $\tau_l = \frac{1}{N-1} \sum_{k \neq l} \mathrm{RMI}(\mathbf{G}_l, \mathbf{G}_k)$
    Form sharing intervals $\mathcal{S}$ greedily ;            `// based on $\tau_l$`
▷ Step 3: Heterogeneity-based Rank Allocation (HRA)
**for** *each interval $[s_k, e_k] \in \mathcal{S}$* **do**
    $H_{\mathrm{interval}_k} = \max_{l \in [s_k, e_k]} H(\mathbf{G}_l)$ ;          `// based on equation 5`
Normalize $\{H_{\mathrm{interval}_k}\}$ to get allocation factors $F_k$ ;    `// based on equation 6`
**for** *each interval k* **do**
    $r_k = \mathrm{round}(F_k \times (N \times r_{vanilla}))$ ;         `// based on equation 7`
**return** $\mathcal{S}, \mathcal{R}$

---

proxy entropy among all layers within that interval:

$$H_{\mathrm{interval_k}} = \max_{l \in [s_k, e_k]} H(\mathbf{G}_l). \tag{5}$$

This approach prevents high-information layers from being "dragged down" by less informative layers within the same shared block, thereby safeguarding the model's critical information flow.

With the representative proxy entropy for each of the $K$ sharing intervals $\{[s_1, e_1], \ldots, [s_K, e_K]\}$, we normalize these values to derive an allocation factor for each interval. This normalization ensures that the sum of factors equals one, allowing for proportional rank distribution. The allocation factor $F_k$ for interval $k$ is given by:

$$F_k = \frac{H_{\mathrm{interval_k}}}{\sum_{i=1}^{K} H_{\mathrm{interval_i}}}. \tag{6}$$

A higher allocation factor $F_k$ signifies that interval $k$ contains more critical information and thus warrants a larger share of the total LoRA rank budget.

To maintain a comparable overall model capacity and ensure a fair comparison with vanilla LoRA, we fix the total LoRA rank budget across the entire model. If a standard LoRA application assigns a fixed rank $r_{\mathrm{vanilla}}$ to each of the $N$ transformer layers, the total rank budget is $N \times r_{\mathrm{vanilla}}$. Our method distributes this budget adaptively. The allocated rank $r_k$ for the shared LoRA adapter corresponding to interval $k$ is calculated as:

$$r_k = \mathrm{round}\left(F_k \times (N \times r_{\mathrm{vanilla}})\right). \tag{7}$$

Here, the round$(\cdot)$ denotes rounding to the nearest integer. The Local Similarity-based Sharing (LSS) primarily reduces parameters by enabling adaptive sharing across similar layers while maintaining model expressiveness. Simultaneously, the Heterogeneity-based Rank Allocation (HRA) improves performance by adaptively allocating higher ranks to more information-rich intervals. This dual adaptive mechanism collectively allows E²LoRA to significantly reduce trainable parameters while preserving or even enhancing overall model performance. Detailed procedures of HRA are provided in Algorithm 3.

### 3.3 INTEGRATION WITH NON-SHARING AND PARAMETER-SHARING LoRA METHODS

The E²LoRA framework seamlessly integrates with both non-sharing and parameter-sharing LoRA approaches. In the non-sharing setting, layer-specific adapters are replaced by shared adapters over adaptive intervals, with ranks dynamically assigned based on proxy entropy. In the parameter-sharing setting, globally shared components ($A$, $B$ or both) are initialized at the maximum possible

Table 1: Performance of fine-tuning Roberta-Base on 5 sub-tasks of the GLUE benchmark.

| Method | #Params | MNLI | SST2 | COLA | QNLI | QQP | Average |
|---|---|---|---|---|---|---|---|
| FFT | 125.00M | 87.60 | 94.80 | 63.60 | 92.80 | 91.90 | 86.14 |
| DoRA (rank4) | 0.17M | 85.39 | 94.61 | 62.35 | 92.48 | 88.98 | 84.76 |
| DoRA | 0.31M | 86.03 | 94.04 | 61.53 | 92.29 | 89.94 | 84.77 |
| AdaLoRA (rank4) | 0.15M | 86.85 | 94.15 | 58.55 | 91.13 | 89.01 | 83.94 |
| AdaLoRA | 0.30M | 87.09 | 94.61 | 58.55 | 91.87 | 89.54 | 84.33 |
| LoRA+ | 0.30M | 87.20 | 94.61 | 63.31 | 92.25 | 90.75 | 85.62 |
| LoRA-FA | 0.15M | 86.80 | 94.80 | 63.60 | 92.50 | 90.10 | 85.56 |
| VeRA | 0.04M | 85.63 | 92.77 | 60.31 | 91.94 | 88.51 | 83.83 |
| LoRA | 0.30M | 87.49 | 94.57 | 63.85 | 92.73 | 90.70 | 85.87 |
| Ours ($E^2$LoRA) | 0.16M | 86.89 | 94.69 | 63.37 | 92.45 | 90.44 | 85.57 |
| ShareLoRA | 0.16M | 86.99 | 94.19 | 61.17 | 92.46 | 90.15 | 84.99 |
| Ours ($E^2$ShareLoRA) | 0.08M | 86.74 | 94.61 | 63.19 | 92.46 | 90.00 | 85.40 |

rank; during computation, only the dimensions corresponding to each layer's dynamically determined rank are accessed. This dynamic slicing mechanism allows $E^2$LoRA to retain the efficiency of existing global sharing strategies while enabling proxy entropy–guided rank allocation.

## 4 EXPERIMENTS

We assess $E^2$LoRA's effectiveness across diverse model architectures and tasks, including natural language understanding, natural language generation, and image classification. To conduct a comprehensive assessment, we integrate $E^2$LoRA with both non-sharing and sharing variants. Specifically, for the non-sharing configuration, we combine it with vanilla LoRA, while for the sharing setting we adopt ShareLoRA, which globally shares the $A$ matrix while retaining layer-specific $B$ matrices. To guarantee fair comparisons, we strictly maintain identical training configurations as those used by both LoRA and ShareLoRA.

### 4.1 NATURAL LANGUAGE UNDERSTANDING (NLU) TASKS

For NLU tasks, we fine-tune the Roberta-Base (Liu et al., 2019) model on five subsets of GLUE benchmarks (Wang et al., 2018), specifically MNLI, SST-2, CoLA, QNLI, and QQP. Detailed training configurations are provided in Appendix C.3.

**Results.** Table 1 summarizes performance on five GLUE tasks. $E^2$LoRA demonstrates superior parameter efficiency, achieving an average score of 85.57 with 0.16M trainable parameters, comparable to LoRA's 85.87 (0.3M parameters). Furthermore, $E^2$ShareLoRA significantly boosts efficiency and effectiveness: it attains 85.40 with just 0.08M parameters, outperforming ShareLoRA's 84.99 (0.16M parameters). This highlights $E^2$LoRA's capability to enhance both model efficiency and accuracy by effectively managing parameter sharing.

### 4.2 NATURAL LANGUAGE GENERATION (NLG) TASKS

For NLG tasks, we evaluate mathematical reasoning and code generation capabilities using a 100K subset of MetaMathQA (Yu et al., 2023) for mathematics and a 100K code-only subset of Code-FeedBack (Zheng et al., 2024) for coding. For evaluation, we employ accuracy scoring on GSM8K (Cobbe et al., 2021) outputs and PASS@1 for HumanEval (Chen et al., 2021). Detailed training configurations are provided in Appendix C.4.

**Results.** Table 2 summarizes $E^2$LoRA's performance and parameter efficiency on GSM8K and HumanEval. $E^2$LoRA, integrated with vanilla LoRA, halves trainable parameters from 6.82M to 3.66M, improving GSM8K accuracy from 70.21 to 70.51 and achieving a 1.63-point gain on HumanEval. $E^2$ShareLoRA further reduces the number of parameters from 2.75M to 1.59M, less than 60% of ShareLoRA's parameters. While $E^2$ShareLoRA's performance on GSM8K remains comparable at 70.26% versus 70.51%, it significantly improves pass@1 on HumanEval from 43.69 to 44.71. This demonstrates $E^2$LoRA's ability to maintain expressiveness and significantly reduce parameter counts through adaptive sharing interval partitioning and rank allocation. To fur-

Table 2: Performance of fine-tuning Llama-3.1-8b-Base on GSM8k and HumanEval.

| Method | GSM8K | | HUMANEVAL | |
| --- | --- | --- | --- | --- |
| | #Params | Acc | #Params | Pass@1 |
| FFT (Attn-only) | 1.34B | 73.69 | 1.34B | 51.63 |
| DoRA (rank4) | 3.74M | 67.40 | 3.74M | 43.90 |
| DoRA | 6.82M | 69.17 | 6.82M | 43.70 |
| AdaLoRA (rank4) | 3.41M | 56.56 | 3.41M | 35.37 |
| AdaLoRA | 6.82M | 70.63 | 6.82M | 41.46 |
| LoRA+ | 6.82M | 71.29 | 6.82M | 44.51 |
| LoRA-FA | 3.41M | 68.66 | 3.41M | 42.07 |
| VeRA | 0.33M | 67.63 | 0.33M | 42.48 |
| LoRA | 6.82M | 70.21 | 6.82M | 42.68 |
| Ours ($E^2$LoRA) | 3.61M | 70.51 | 3.66M | 44.31 |
| ShareLoRA | 2.75M | 70.51 | 2.75M | 43.69 |
| Ours ($E^2$ShareLoRA) | 1.59M | 70.26 | 1.59M | 44.71 |

Table 3: Performance of fine-tuning CLIP-ViT-B/16 across multiple image classification datasets.

| Method | #Params | Cars | DTD | Eurosat | GTSRB | Resisc45 | Sun397 | SVHN | Average |
| --- | --- | --- | --- | --- | --- | --- | --- | --- | --- |
| Zero-shot | - | 63.75 | 44.39 | 42.22 | 35.22 | 56.46 | 62.56 | 15.53 | 45.73 |
| FFT (ViT-only) | 86.38M | 84.23 | 77.44 | 98.09 | 94.31 | 93.95 | 75.35 | 93.04 | 88.06 |
| DoRA (rank=4) | 0.75M | 81.43 | 74.79 | 98.31 | 95.68 | 94.06 | 76.41 | 96.37 | 88.15 |
| DoRA | 1.39M | 82.44 | 76.86 | 98.43 | 97.25 | 95.10 | 77.30 | 96.63 | 89.14 |
| AdaLoRA (rank=4) | 0.66M | 72.35 | 72.18 | 96.46 | 44.16 | 87.00 | 70.80 | 93.49 | 76.63 |
| AdaLoRA | 1.31M | 73.29 | 74.31 | 97.06 | 58.87 | 88.51 | 71.76 | 94.21 | 79.72 |
| LoRA+ | 1.31M | 86.61 | 73.33 | 98.54 | 98.99 | 96.06 | 76.80 | 96.98 | 89.62 |
| LoRA-FA | 0.66M | 79.55 | 73.78 | 98.23 | 92.84 | 93.85 | 74.76 | 95.96 | 86.98 |
| VeRA | 0.10M | 82.36 | 76.35 | 98.29 | 97.33 | 95.21 | 76.80 | 96.48 | 88.97 |
| LoRA | 1.31M | 82.26 | 76.88 | 98.42 | 97.10 | 95.02 | 77.23 | 96.62 | 89.08 |
| Ours ($E^2$LoRA) | 0.66M | 83.47 | 77.39 | 98.59 | 98.67 | 95.58 | 78.17 | 96.83 | 89.81 |
| ShareLoRA | 0.72M | 82.00 | 75.83 | 98.35 | 96.91 | 94.86 | 76.83 | 96.47 | 88.75 |
| Ours ($E^2$ShareLoRA) | 0.39M | 83.27 | 77.48 | 98.50 | 98.40 | 95.62 | 78.02 | 96.79 | 89.73 |

ther verify the method's generality on diverse reasoning tasks, we also evaluated $E^2$LoRA on the CommonSenseQA benchmark. As detailed in Appendix K, $E^2$LoRA maintains a strong efficiency-performance balance even on complex commonsense reasoning subtasks.

### 4.3 IMAGE CLASSIFICATION TASKS

For image classification tasks, we fine-tune the CLIP-ViT-B/16 model on seven widely used image classification benchmarks. Detailed training configurations are provided in Appendix C.5.

**Results.** Table 3 summarizes $E^2$LoRA's robust performance and parameter efficiency across image classification tasks. $E^2$LoRA consistently achieves superior or comparable accuracy to vanilla LoRA while utilizing approximately half its trainable parameters, 0.66M versus 1.31M. This yields an average accuracy improvement of 0.73, reaching 89.81%, with significant gains on GTSRB and Cars. Integrating with ShareLoRA, $E^2$ShareLoRA further enhances efficiency, reducing the number of parameters from 0.72M to 0.39M. Simultaneously, it boosts average accuracy by around 0.98, reaching 89.73%. Substantial improvements are observed across all datasets. This consistent uplift underscores $E^2$LoRA's ability to mitigate performance decline in globally shared methods, optimizing both efficiency and accuracy.

## 5 DISCUSSION

**Effect of LSS and HRA.** To validate the effectiveness of our proposed components, LSS and HRA, we conducted an ablation study, the results of which are summarized in Table 4. LSS alone significantly reduces trainable parameters while maintaining competitive accuracy, demonstrating its efficacy in parameter reduction through adaptive sharing. HRA notably enhances model performance

by optimally distributing parameters based on layer-wise information heterogeneity. Together, these components synergistically enhance both parameter efficiency and performance.

Table 4: Ablation Study of Different Components of $E^2$LoRA, evaluated using Llama3.1-8B on GSM8K.

| LSS | HRA | #Params | ACC |
|---|---|---|---|
| ✓ | ✗ | 3.51M | 69.45 |
| ✗ | ✓ | 6.82M | 71.19 |
| cosine | ✓ | 3.75M | 66.67 |
| L1 | ✓ | 3.75M | 68.48 |
| L2 | ✓ | 3.75M | 67.93 |
| KL | ✓ | 3.43M | 67.57 |
| ✓ | ✓ | 3.61M | 70.51 |

Table 5: Compare $E^2$LoRA and LoRA with similar amounts of parameters by adjusting rank, evaluated using Llama3.1-8B on GSM8K.

| | Rank | #Params | Acc |
|---|---|---|---|
| LoRA | 4 | 3.41M | 67.58 |
| | 8 | 6.82M | 70.21 |
| $E^2$LoRA | 8 | 3.61M | 70.51 |
| | 16 | 7.22M | 71.19 |

**Efficiency Comparison at Matching Parameters.** To further assess the parameter efficiency of $E^2$LoRA compared to LoRA, we conduct an experiment where we control the rank to ensure similar parameter counts. As shown in Table 5, under comparable parameter budgets, $E^2$LoRA consistently outperforms LoRA with higher ranks. Specifically, $E^2$LoRA with rank 8 surpasses LoRA with rank 4 by 0.3 points. Moreover, $E^2$LoRA with rank 16 outperforms LoRA with rank 8 by a margin of 0.98 points. These results confirm that with the same parameter scale, $E^2$LoRA also achieves better performance, further demonstrating its superior parameter efficiency. We further extend this parameter-matched comparison to Computer Vision (CV) and Natural Language Understanding (NLU) tasks. As detailed in Appendix J, consistent improvements are observed across all modalities, verifying the generalizability of our efficiency gains.

**Orthogonality with Existing LoRA Variants.** Building on our experiments where $E^2$LoRA augments vanilla LoRA and ShareLoRA, we further test its compatibility with diverse PEFT variants. $E^2$LoRA acts as a plug-and-play enhancement guided by intrinsic properties of the pre-trained model, and is thus orthogonal to the particular adapter architecture. We apply its adaptive sharing and rank allocation to performance-oriented DoRA and efficiency-focused VeRA and LoRI; as shown in Table 6, the resulting E²DoRA, E²VeRA, and E²LoRI consistently improve parameter efficiency, with E²DoRA matching or

Table 6: Demonstration of $E^2$LoRA's orthogonality. By applying our adaptive framework to existing LoRA variants (DoRA, VeRA, LoRI), we consistently achieve significant parameter reduction while maintaining or improving performance across GLUE (NLU) and a collection of CV datasets. 'E²X' denotes the combination of $E^2$LoRA with method X.

| Method | NLU (GLUE) | | CV | |
|---|---|---|---|---|
| | #Params | Avg. | #Params | Avg. |
| DoRA | 0.31M | 84.77 | 1.39M | 89.14 |
| $E^2$DoRA | 0.15M | 84.79 | 0.75M | 89.82 |
| VeRA | 0.04M | 83.83 | 0.10M | 88.97 |
| $E^2$VeRA | 0.013M | 84.31 | 0.05M | 89.13 |
| LoRI-S | 0.015M | 78.26 | 0.066M | 89.47 |
| $E^2$LoRI-S | 0.007M | 78.96 | 0.033M | 89.43 |

surpassing DoRA on NLU, CV, and NLG tasks using about 50% fewer parameters, E²VeRA and E²LoRI reaching comparable or better accuracy while further reducing parameter counts. These results support $E^2$LoRA as a general-purpose compression strategy that amplifies the efficiency of existing and future LoRA-style methods.

**Effect of Different Similarity Metrics.** An effective parameter sharing strategy hinges critically on accurately quantifying inter-layer similarity to enable adaptive sharing ranges. As demonstrated in Table 4 and Figure 3, our proposed proxy entropy similarity metric proves highly effective for this purpose. Specifically, Figure 3(a) reveals that proxy entropy similarity captures fine-grained local dependencies essential for precise adaptive range partitioning. In stark contrast, Figure 3(b) shows that cosine similarity produces a nearly uniform and uninformative similarity matrix, rendering it ill-suited for guiding adaptive parameter sharing. This observation is further corroborated quantitatively: substituting our entropy-based similarity with other similarity metrics leads to a significant performance drop, highlighting the indispensable role of entropy-based similarity in achieving both effective and performance-preserving parameter sharing. Additional visual comparisons of various similarity metrics are provided in Appendix D.3.

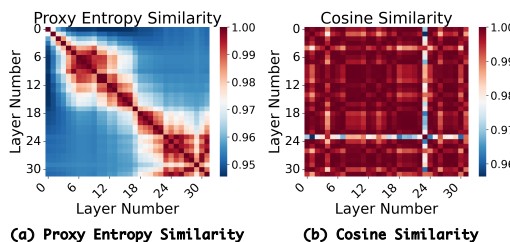

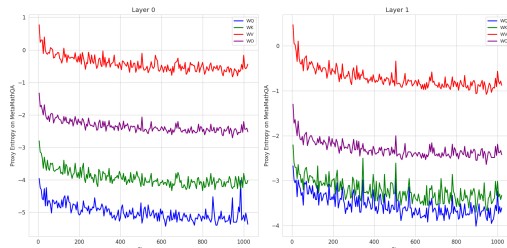

Figure 3: Visual comparison of different similarity metrics with Llama-3.1-8B trained on the MetaMathQA dataset. Visualization of other metrics can be found in the Appendix D.3.

Figure 4: Gradient proxy entropy dynamics across training for attention matrices ($W_Q$, $W_K$, $W_V$, $W_O$) in layers 0,1 of Llama-3.1-8B trained on MetaMathQA.

**Step-wise Importance Analysis.** We conduct a step-wise importance analysis during fine-tuning and find that module importance remains stable across steps. $W_V$ modules dominate $W_K$ and $W_Q$, and middle layers are more important than upper layers, consistent with prior research. This stability justifies our static rank allocation based on initial gradient entropy without extra computation. Figure 4 and Appendix E report representative dynamics and further analyses.

**Effect of rank.** As shown in the left panel of Figure 5, scaling the rank consistently improves performance on both datasets. Importantly, our method not only maintains competitive performance at lower ranks but also yields substantial performance gains with higher ranks.

**Effect of step.** The sampling step controls the data used for gradient computation with Subset Size = Steps × Batch Size. The right panel of Figure 5 shows that performance

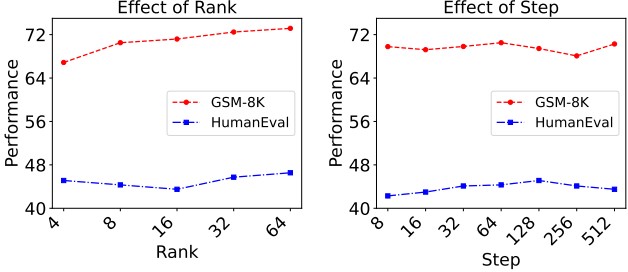

Figure 5: Ablation study of rank and step size, evaluated using Llama-3.1-8B on GSM8K and HumanEval.

is stable across a wide range of step values, indicating our method is robust to the choice of step.

**Robustness Analysis.** We evaluate the stability of our partitioning and rank allocation under different random seeds to reflect sampling variability, as reported in Appendix I. We also examine the robustness of RMI-based thresholding under noise perturbations, with results presented in Appendix H. These results show that our adaptive decisions remain consistent and insensitive to small estimation noise, demonstrating the structural stability and practical reliability of the framework.

**Experiments on larger and diverse models.** We further evaluate $E^2$LoRA on Llama-2-13B, Qwen3-8B, and GPT-OSS-20B and observe that it consistently outperforms standard LoRA with comparable accuracy and much fewer trainable parameters, as summarized in Appendix L.

**Computation Time and Reusability.** The initialization overhead is negligible, requiring approximately 30 seconds compared to 40 minutes for fine-tuning. Crucially, the determined sharing configurations can be reused for the same model and task, rendering this a one-time cost when exploring different training setups, such as hyperparameter tuning.

## 6 CONCLUSION

In this paper, we propose $E^2$LoRA, a novel parameter- sharing framework for efficient and effective Low-Rank Adaptation. Inspired by a gradient entropy-based analysis of pre-trained models, we introduced local similarity-based shared area division and layer-wise heterogeneity-based rank allocation. $E^2$LoRA effectively halves the number of trainable parameters while consistently maintaining or even surpassing the original performance. Extensive validation confirms the effectiveness and robust generalizability of $E^2$LoRA across diverse model architectures and datasets, alongside its seamless integration with existing sharing and non-sharing LoRA variants across every task.

## CONTRIBUTION

Minglei Li: Investigation, Methodology, Software, Visualization, Writing – original draft. Peng Ye: Methodology, Formal analysis, Conceptualization, Visualization, Writing – review & editing. Jingqi Ye: Methodology, Validation, Writing – review & editing. Haonan He: Software, Validation, Writing – review & editing. Tao Chen: Supervision, Resources, Project administration.

## ACKNOWLEDGMENTS

This work is supported by National Key Research and Development Program of China (No. 2022ZD0160101), Shanghai Natural Science Foundation (No. 23ZR1402900), Shanghai Science and Technology Commission Explorer Program Project (24TS1401300), Shanghai Municipal Science and Technology Major Project (No.2021SHZDZX0103). The computations in this research were performed using the CFFF platform of Fudan University.

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

## A   NOTATION

| Symbol | Description |
|---|---|
| $N$ | The total number of layers in the pre-trained model. |
| $G_l$ | The gradient tensor for the $l$-th layer. |
| $RMI(G_i, G_j)$ | Relative Mutual Information between layer $i$ and $j$, normalized to $[0, 1]$ to measure similarity. |
| $\mathcal{S} = \{[s_k, e_k]\}$ | The set of adaptive sharing intervals, where $s_k$ is the start layer index and $e_k$ is the end layer index of the $k$-th interval. |
| $H_{interval_k}$ | The representative entropy for an interval, defined as the **maximum** proxy entropy among all layers within that interval $(\max_{l \in [s_k, e_k]} H(G_l))$. |
| $F_k$ | The normalised allocation factor for interval $k$, used to distribute the rank budget. |
| $r_{vanilla}$ | The base rank used in standard LoRA (e.g., 8), used to calculate the total rank budget. |

Table 7: Notation Table

## B   THEORETICAL JUSTIFICATION OF PROXY ENTROPY

Below we make explicit (i) what the proposed proxy entropy actually measures, and (ii) how it connects, in a mathematically rigorous way, to standard information-theoretic quantities.

In summary, the proposed proxy entropy is *not* a heuristic approximation to Shannon entropy, but a mathematically well-defined quantity that is *exactly* a monotone transform of the per-parameter Frobenius norm of the gradient. Since the gradient Frobenius norm is a standard Monte Carlo estimator of the per-layer Fisher-information trace, a classical information-theoretic measure of parameter sensitivity, our proxy entropy inherits a direct and rigorous connection to established information-theoretic principles. The following paragraphs provide the detailed derivation supporting this conclusion.

**(1) From proxy entropy to Frobenius norm.**   For layer $\ell$, let $g_\ell \in \mathbb{R}^{d_\ell}$ be the flattened gradient and $\mu_\ell$ its mean. The sample variance is

$$\sigma_\ell^2 = \frac{1}{d_\ell} \sum_{i=1}^{d_\ell} (g_{\ell,i} - \mu_\ell)^2.$$

Let $\tilde{G}_\ell$ be the centered gradient tensor. Then we have

$$\|\tilde{G}_\ell\|_F^2 = \sum_{i=1}^{d_\ell} (g_{\ell,i} - \mu_\ell)^2 = d_\ell \sigma_\ell^2,$$

then

$$\log \sigma_\ell = \log \|\tilde{G}_\ell\|_F - \tfrac{1}{2} \log d_\ell.$$

Our proxy entropy is

$$H(G_\ell) = \log(\sigma_\ell) + \tfrac{1}{2} \log(2\pi) + \tfrac{1}{2},$$

so $H(G_\ell)$ is exactly a monotone transform of the *per-parameter Frobenius norm* of the centered gradient. Thus, in purely mathematical terms, our metric is not an ad-hoc heuristic but a normalized, log-compressed gradient Frobenius norm.

**(2) Frobenius norm and Fisher information.**   In standard settings where the gradient corresponds to the score function of a loss or log-likelihood, the Fisher information matrix per-layer is

$$F_\ell = \mathbb{E}\big[g_\ell g_\ell^\top\big],$$

and its trace is

$$\mathrm{tr}(F_\ell) = \mathbb{E}\big[\|g_\ell\|_2^2\big].$$

Hence, the squared Frobenius norm of the gradient is an unbiased empirical estimator of $\mathrm{tr}(F_\ell)$, i.e. of the *total Fisher information* at layer $\ell$. Our proxy entropy $H(G_\ell)$, being a monotone transform of $\|\tilde{G}_\ell\|_F$, is therefore also a monotone transform of an empirical estimate of the per-layer Fisher trace.

Fisher information measures how sensitive the model is to small changes of the parameters: layers with larger Fisher information are more "informative" or important for the task. Since our proxy entropy $H(G_\ell)$ is a monotone transform of the per-parameter Frobenius norm of the gradient, it can be viewed as a scalar summary of how much Fisher information each layer carries, which ties it directly to a classical information-theoretic notion in a simple and practical way.

# C DETAILED EXPERIMENT SETTINGS

## C.1 RANDOM SEED CONTROL.

**It is important to note that all experimental results are averaged over three independent runs**, initialized with distinct random seeds, to ensure the robustness and reproducibility of our findings.

## C.2 BASELINE METHODS.

We compare E$^2$LoRA with a range of established parameter-efficient fine-tuning methods, categorized by their primary objective and sharing mechanism. This comprehensive comparison highlights the effectiveness and efficiency of our approach.

- **Non-Sharing Methods:** These methods typically adapt models without explicit parameter sharing across layers.
    - **LoRA** (Hu et al., 2021): The vanilla Low-Rank Adaptation method, which injects trainable low-rank matrices into pre-trained models to preserve performance with minimal additional parameters.
    - **DoRA** (Liu et al., 2024a): An effectiveness-enhancing variant that re-parameterizes LoRA by decomposing updates to pre-trained weights into magnitude and direction components.
    - **LoRA+** (Hayou et al., 2024): An effectiveness-boosting approach that optimizes LoRA's learning dynamics through imbalance learning rates.
    - **AdaLoRA** (Zhang et al., 2023c): An effectiveness-enhancing variant that dynamically adjusts the rank budget for LoRA adapters during training.
    - **LoRA-FA** (Zhang et al., 2023b): An efficiency-enhancing method that freezes the A matrices of LoRA adapters, training only the B matrices to reduce trainable parameters.
- **Sharing Methods:** These methods explicitly incorporate parameter sharing mechanisms to enhance efficiency.
    - **ShareLoRA** (Song et al., 2024): An efficiency-boosting variant that shares the A matrix across all layers while maintaining layer-specific B matrices.
    - **VeRA** (Kopiczko et al., 2023): An efficiency-oriented method that implements a constrained approach by sharing fixed low-rank matrices across layers and updating only introduced layer-specific scaling vectors.

## C.3 DETAILED SETTINGS OF NLU.

We adopt the hyperparameters used in LoRA (Hu et al., 2021). These same hyperparameters are consistently applied across LoRA, ShareLoRA, and combinations of our proposed method with them. We use the Adam (Kingma & Ba, 2014) optimizer ($\beta_1 = 0.9$, $\beta_2 = 0.999$, weight decay=0.1) with a cosine learning rate scheduler (6% warmup). LoRA is applied only to the query and value projections. Task-specific hyperparameters (peak learning rates, batch sizes) align with LoRA and ShareLoRA. All experiments use FP32 precision and a maximum sequence length of 512 tokens. The sampling step of our method is 64. Other details are shown in Table 8.

Table 8: Hyperparameters for fine-tuning RoBERTa-base LoRA on GLUE benchmarks.

| Parameter | MNLI | SST-2 | CoLA | QNLI | QQP |
|---|---|---|---|---|---|
| Optimizer | | | AdamW | | |
| Warmup Ratio | | | 0.06 | | |
| LR Schedule | | | Linear | | |
| Batch Size | 16 | 16 | 32 | 32 | 16 |
| # Epochs | 30 | 60 | 80 | 25 | 25 |
| Learning Rate | 5E-04 | 5E-04 | 4E-04 | 4E-04 | 5E-04 |
| LoRA Config. | | | $r_q = r_v = 8$ | | |
| LoRA $\alpha$ | | | 16 | | |
| Max Seq. Len. | | | 512 | | |

## C.4 DETAILED SETTINGS OF NLG.

For experiments on Llama3.1-8B-Base, including both baseline methods and $E^2$LoRA, we use one of the most commonly adopted hyperparameter configurations, unless stated otherwise in the main text. We use the AdamW (Loshchilov & Hutter, 2017) optimizer ($\beta_1 = 0.9$, $\beta_2 = 0.999$, weight decay = 5e-4, batch size = 64) with a cosine learning rate schedule (3% warmup). We use the BF16 mixed precision strategy (Kalamkar et al., 2019) and apply LoRA to linear layers in attention modules. The peak learning rates are configured as follows: 5e-5 for our method and standard baselines (LoRA/ShareLoRA), 5e-4 for AdaLoRA and LoRA-FA, and 5e-3 for VeRA to achieve meaningful results. Other details are shown in Table 9.

## C.5 DETAILED SETTINGS OF IMAGE CLASSIFICATION TASKS.

For image classification tasks, following LoRA-Pro (Wang et al., 2024b), we fine-tune CLIP-ViT-B/16 on StanfordCars (Krause et al., 2013), DTD (Krause et al., 2013), EuroSAT (Helber et al., 2019), GTSRB (Houben et al., 2013), RESISC45 (Cheng et al., 2017), SUN397 (Xiao et al., 2010), and SVHN (Netzer et al., 2011) with accuracy used as the primary metric. During fine-tuning, only the linear layers of the visual backbone are updated using FP32 precision. The classification result is derived from prompts such as "a photo of a {class}." Other details are shown in Table 9.

Table 9: Hyper-parameters used in experiments

| Hyperparameter | Llama3.1-8B-Base | CLIP-ViT-B/16 |
|---|---|---|
| LR | 5e-5 | 1e-4 |
| LR Decay | 0.1 | 0 |
| Warmup | 0.3 | 0.3 |
| Optimizer | AdamW | Adam |
| Betas | 0.9, 0.999 | 0.9, 0.999 |
| Weight Decay | 5e-4 | 0 |
| Batch Size | 64 | 64 |

## C.6 TRAINING ENVIRONMENT.

All codes of $E^2$LoRA and baseline methods are implemented in PyTorch. For natural language understanding and image classification tasks, we conduct our experiments on a single A800 GPU per task, using the Huggingface Transformers framework for model and trainer implementation. For natural language generation tasks, we conduct experiments on 8 A800 GPUs, utilizing the Deep-Speed ZeRO2 (Rajbhandari et al., 2020) data parallel framework and FlashAttention-2 (Dao, 2023) mechanism.

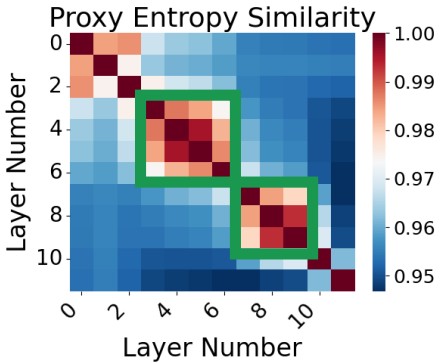 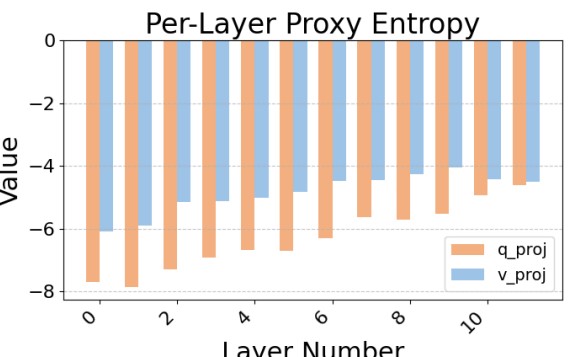

Figure 6: Gradient-based proxy entropy analysis for CLIP-ViT-B/16 on Cars. Left panels show inter-layer information similarity, with high-similarity blocks highlighted in green. Right panels illustrate absolute proxy entropy values across layers, indicating significant variations.

## D ADDITIONAL FIGURES AND VISUALIZATIONS

### D.1 EXTENDED FIGURES OF GRADIENT-BASED PROXY ENTROPY ANALYSIS

Figure 6 presents the gradient-based proxy entropy analysis for CLIP-ViT-B/16 on the Cars dataset, which further corroborates our two key findings: *Local Similarity* and *Layer-wise Information Heterogeneity*.

### D.2 UNIVERSALITY ACROSS DIVERSE LLM ARCHITECTURES

To rigorously validate the universality of our observations, we extended the gradient-based proxy entropy analysis to a broader range of Large Language Models (LLMs) with varying scales and architectures on GSM8K, a benchmark of math reasoning. As shown in Figure 7, we visualized the layer-wise gradient similarity and entropy for **Qwen3-8B**, **Phi4-mini**, **GPT-OSS-20B** (a Mixture-of-Experts model), and **Gemma-9B-it**, utilizing gradients derived from the GSM8K training set.

The results align perfectly with our initial findings in Figure 1 and the CLIP analysis above. Specifically:

- **Local Similarity:** The heatmaps (left panels for each model) consistently exhibit distinct diagonal block patterns across all models, regardless of whether they are Dense or MoE-based. This confirms that adjacent layers tend to share similar gradient information during math reasoning tasks.

- **Information Heterogeneity:** The bar charts (right panels) display significant fluctuations in proxy entropy values across layers, verifying that different layers contribute unevenly to the learning process on GSM8K.

These consistent patterns across diverse model families, scales, and structures strongly support the generalizability of the underlying principles driving $E^2$LoRA.

### D.3 EXTENDED VISUALIZATIONS OF VARIOUS SIMILARITIES

Figure 8 presents comparative visualizations of alternative similarity metrics. Panels (a), (c), and (d), depicting Cosine, L1, and L2 similarities respectively, reveal largely featureless or uniformly high similarity matrices. This lack of discernible local similarity patterns visually confirms their ineffectiveness for precise adaptive region partitioning, aligning with the observed performance degradation in Table 4. While L1 and L2 similarities exhibit structural resemblances due to their computational definitions, their distinct magnitudes can lead to varying impacts on subsequent processes, particularly in rank allocation, thereby explaining their differing performances observed in Table 4. Panel (b) shows KL similarity, which does indicate local similarity regions. However, these

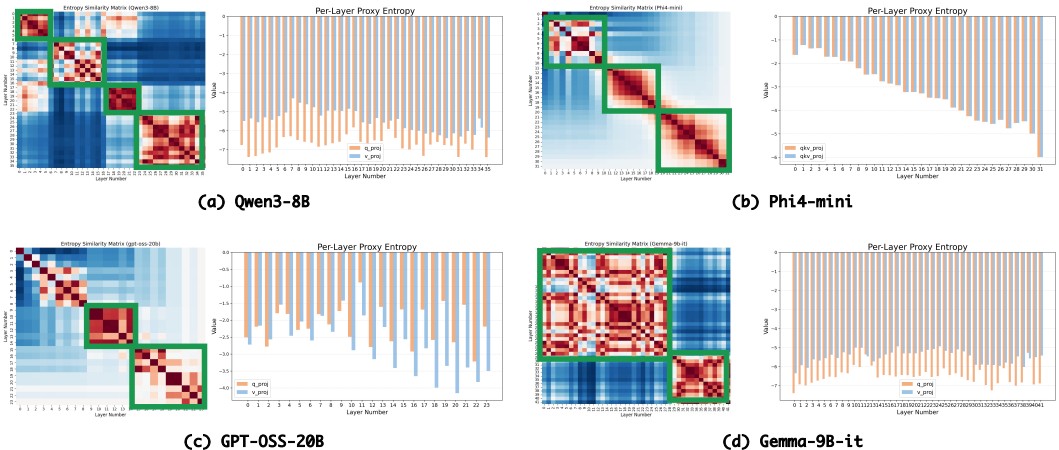

Figure 7: Gradient-based proxy entropy analysis on diverse LLMs **on the GSM8K dataset**: (a) Qwen3-8B, (b) Phi4-mini, (c) GPT-OSS-20B (MoE), and (d) Gemma-9B-it. Consistent with our main observations, all models exhibit clear local similarity patterns (diagonal blocks in heatmaps) and significant layer-wise entropy heterogeneity (varying bar heights) when fine-tuned on math reasoning tasks.

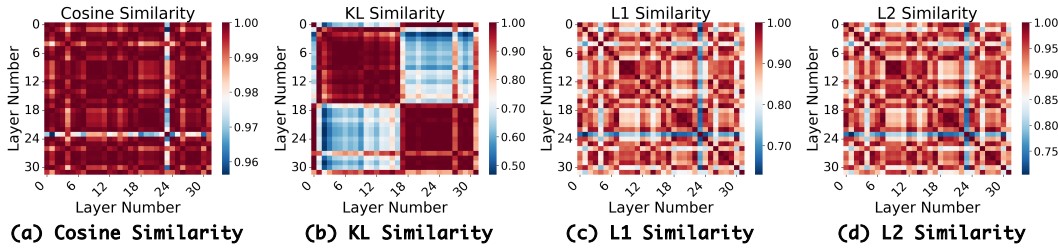

Figure 8: Visual comparison of various similarity measures. Panels (a), (b), (c), and (d) display Cosine, KL, L1, and L2 similarities, respectively.

regions are often excessively broad, hindering the identification of genuinely proximate and informationally relevant layers for fine-grained adaptive sharing. This contrasts sharply with our proxy entropy similarity, which provides the necessary granularity for effective parameter sharing.

## E   DETAILED STEP-WISE IMPORTANCE ANALYSIS

To assess whether the importance of targeted modules varies significantly during fine-tuning, we conduct an experiment using full fine-tuning of Llama-3.1-8B-Base (Grattafiori et al., 2024) on MetaMathQA (Yu et al., 2023). We compute the gradient entropy for each weight matrix every 5 optimization steps to quantify module-wise importance dynamics. For optimization, we employ AdaLomo (Lv et al., 2023), a layer-wise, state-free, SGD-like optimizer with adaptive learning rates, configured with a standard large learning rate (1e-4) and run for 1,024 steps.

As illustrated in Figure 9-12, the relative importance of targeted modules remains remarkably stable throughout fine-tuning. Notably, across nearly all layers, WV modules dominate in importance, while WK and WQ modules consistently exhibit lower influence—a finding aligned with prior work demonstrating that fine-tuning WV yields superior performance compared to fine-tuning other modules (Hu et al., 2021; Zhang et al., 2023c; He et al., 2025a).

To further analyze layer-wise trends, we aggregate the entropy of targeted modules within each layer (Figure 13). The results reveal stable inter-layer importance patterns: middle layers (green/blue) maintain significantly higher importance, while upper layers are consistently least influential. This suggests a consistent hierarchy in gradient-driven learning across depths.

These observations directly justify the design rationale of our method. The persistent disparity in module importance—coupled with its temporal stability during training—validates our approach of static rank allocation based on initial gradient entropy. Specifically, we assign higher ranks to modules exhibiting greater importance (e.g., WQ matrices) as measured before the formal training process. Crucially, this allocation:

- Preserves parameter efficiency: Rank assignments are fixed after initialization, requiring no dynamic shape adjustment.
- Incurs zero computational overhead: Unlike adaptive methods, our strategy avoids runtime reconfiguration costs.

This design is empirically grounded in the observed hierarchy (WV/WO > WK/WQ) and layer-wise stability (middle > upper/lower layers), ensuring optimal resource distribution aligns with inherent module roles.

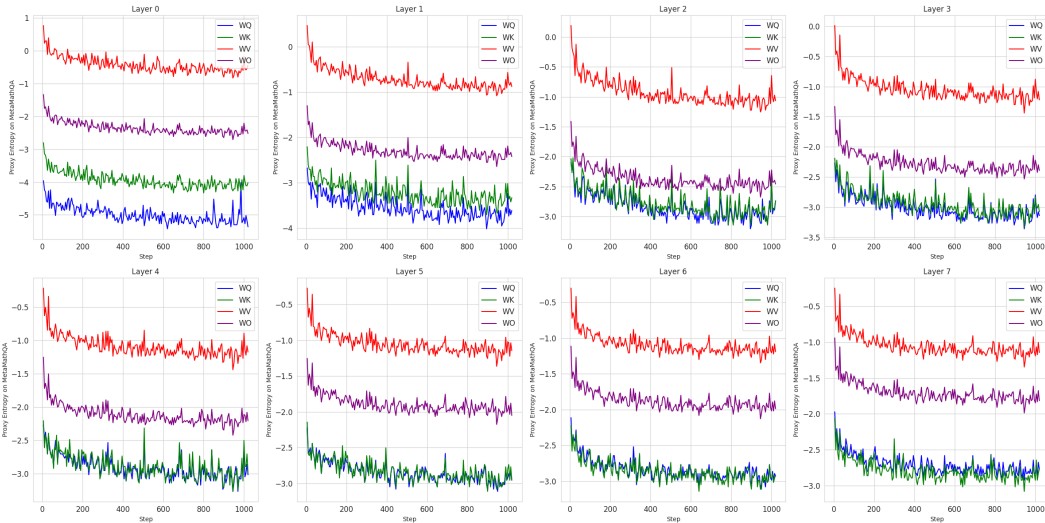

Figure 9: Gradient entropy dynamics across training for attention matrices (WQ, WK, WV, WO) in layers 0-7 of Llama-3.1-8B trained on MetaMathQA. Each subplot shows the evolution of absolute gradient entropy for one layer, revealing distinct patterns across different attention components.

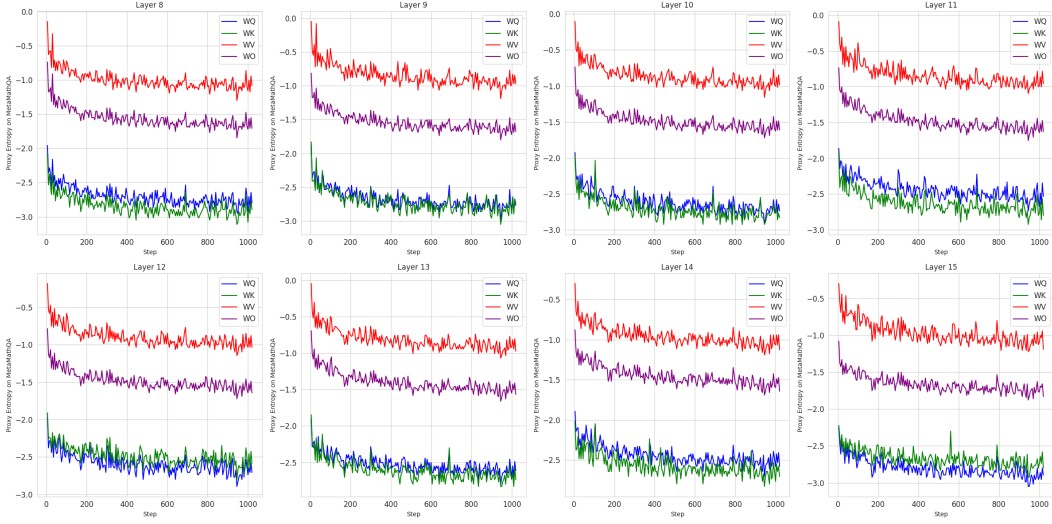

Figure 10: Gradient entropy dynamics for attention matrices (WQ, WK, WV, WO) in layers 8-15, continuing from Figure 9.

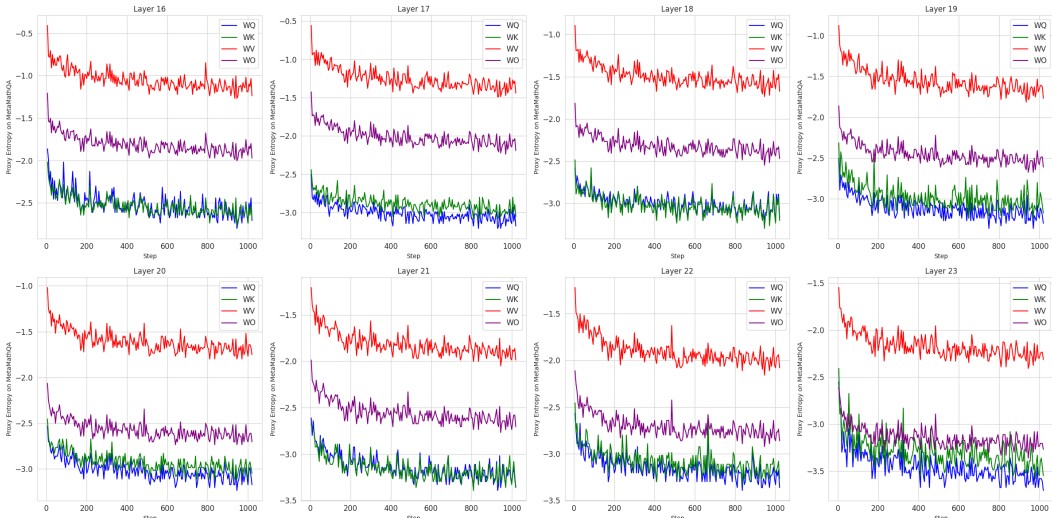

Figure 11: Gradient entropy dynamics for attention matrices in layers 16-23.

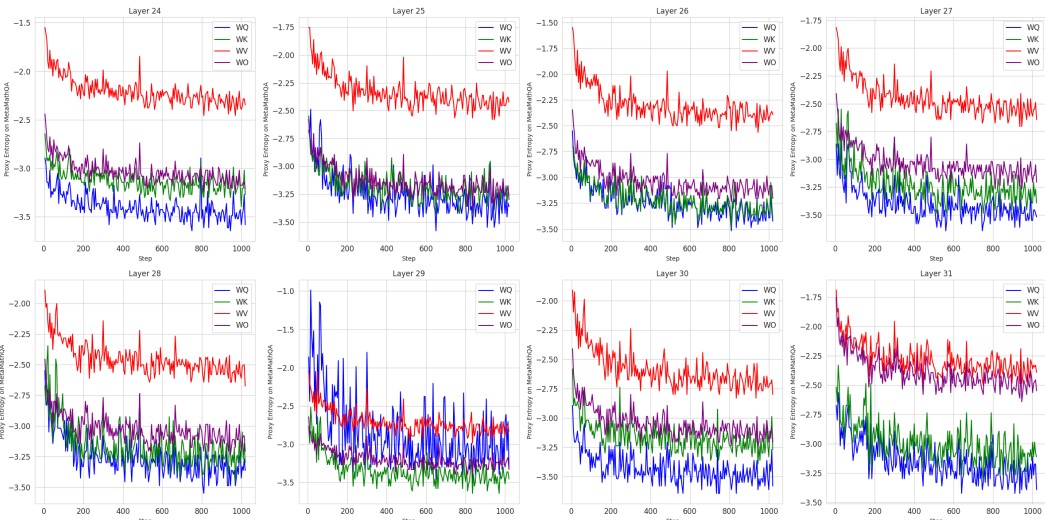

Figure 12: Gradient entropy dynamics for attention matrices in the deepest layers 24-31.

## F  ANALYSIS ON GRADIENT CALCULATION

Our method initiates by performing an initial gradient computation step on a subset of data to identify crucial parameters. The efficiency and memory footprint of this process are critical, and our implementation is designed for minimal overhead.

- **Memory-Efficient Initialization:** To mitigate memory consumption during this initial phase, we employ a layer-wise gradient computation strategy. Gradients for each layer are computed sequentially, followed by immediate aggregation via all-reduce operations and subsequent offloading to CPU memory by the local master process. This approach eliminates the necessity of simultaneously storing full-model gradients and optimizer states on the GPU, ensuring that GPU memory usage remains at levels comparable to inference, primarily occupied by model weights and activations.

- **Absence of Optimizer States:** A significant advantage of our initialization procedure is that it does not require the instantiation or storage of any optimizer states. This contrasts sharply with typical full fine-tuning processes which necessitate considerable memory for

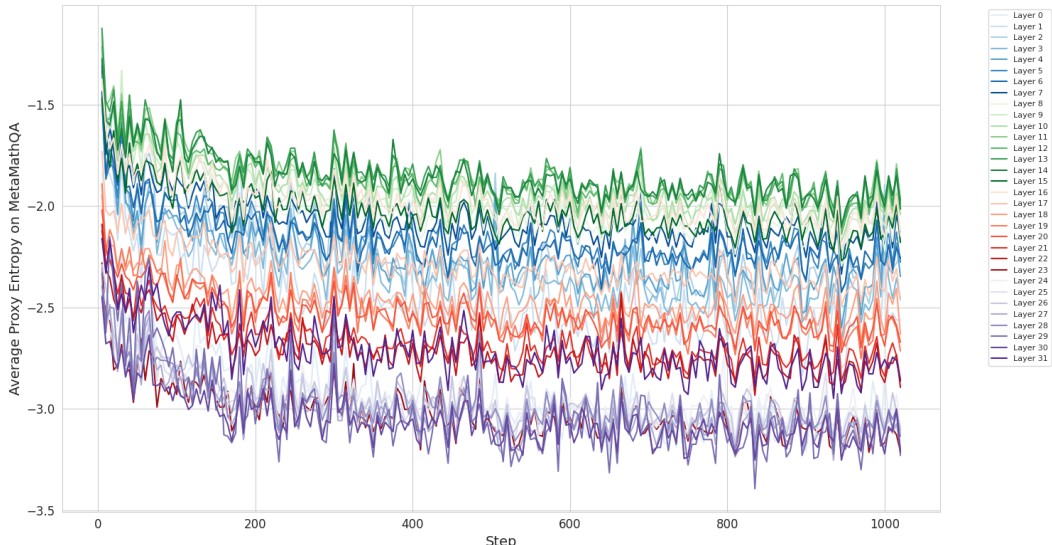

Figure 13: Average gradient entropy across training steps for all 32 layers, grouped into four color-coded sets (layers 0–7: blue; 8–15: green; 16–23: red; 24–31: purple). Each line represents the mean entropy of the four attention weight matrices (WQ, WK, WV, WO) within a layer.

      optimizer states like momentum and variance, leading to substantial memory savings during this critical initial phase.

- **GPU Peak Memory Footprint:** Our initialization process introduces no additional GPU peak memory beyond what is required for standard inference. The peak memory during this phase is primarily comprised of 16-bit model parameters, 16-bit activations, and the gradients for a single layer at a time. This configuration results in significantly lower memory consumption compared to full training, even when advanced optimization techniques like ZeRO-2 are employed during the training phase. For further GPU memory reduction, the entire model can be optionally offloaded to the CPU during this initialization step. While this involves storing 16-bit gradients of targeted modules on the CPU, which incurs additional CPU memory overhead compared to a naive LoRA setup, this is a manageable trade-off given that CPU memory is substantially more affordable and abundant than GPU memory.

## G    VALIDATION OF THE GREEDY STRATEGY VIA GLOBAL OPTIMIZATION

To rigorously validate the effectiveness of our proposed greedy segmentation strategy, we implemented a global optimization baseline based on Dynamic Programming (DP). This comparison aims to determine whether the local decision-making process of the greedy approach compromises global optimality or downstream performance.

### G.1    GLOBAL OPTIMIZATION FORMULATION

We designed a global optimization algorithm that operates on the same Relative Mutual Information (RMI) matrix used by our greedy method. The formulation consists of three steps:

- **Segment Cost Calculation:** For every possible contiguous interval $[i, j]$, we compute a cost $C(i, j)$, defined as the negative mean similarity of layers within that interval based on the RMI matrix.
- **Global Objective:** The total objective function $\mathcal{L}$ is defined as the sum of segment costs plus a regularization term to control granularity:

$$\mathcal{L} = \sum_{k=1}^{K} C(i_k, j_k) + \lambda \cdot K, \tag{8}$$

where $K$ represents the number of intervals and $\lambda$ is a penalty coefficient for over-segmentation.

- **DP Search:** We employ a Dynamic Programming algorithm to strictly minimize $\mathcal{L}$, backtracking to identify the globally optimal partition $\mathcal{S}^*$ for each module type.

## G.2 EXPERIMENTAL RESULTS AND ANALYSIS

We conducted a direct comparison between the globally optimal solution (DP) and our greedy approach on GLUE (RoBERTa) and CV (CLIP-ViT) tasks using identical training setups. The comparative results are summarized in Table 10.

Table 10: Comparison between the Global Optimal (DP) strategy and our Greedy approach. *Time* denotes the search latency during the grouping phase. Results are averaged across datasets.

| Task | Metric | Global Optimal (DP) | Greedy (Ours) |
|---|---|---|---|
| CV (CLIP-ViT) | Search Time | 0.6 ms | **0.2 ms** |
| | Parameters | 0.66 M | 0.66 M |
| | Performance | 89.92 | 89.81 |
| GLUE (RoBERTa) | Search Time | 0.6 ms | **0.2 ms** |
| | Parameters | 0.15 M | 0.16 M |
| | Performance | 85.43 | **85.57** |

**Performance Parity.** As evidenced in Table 10, the global optimal grouping does not yield statistically significant improvements over our greedy approach. Specifically, the DP method shows a marginal gain on CV (+0.11%) but a slight drop on GLUE (-0.14%). This suggests that the local similarity structure of Large Language Models (LLMs) is distinct enough that a greedy strategy effectively captures the necessary boundaries without requiring global look-ahead.

**Efficiency and Robustness.** The greedy approach demonstrates superior efficiency, being approximately $3\times$ faster in search time (0.2 ms vs. 0.6 ms). More critically, the DP approach introduces an additional hyperparameter, the penalty coefficient $\lambda$, which requires tuning. In contrast, our greedy method relies on a self-adaptive threshold derived directly from RMI statistics.

Given that the greedy approach offers comparable performance with lower computational complexity and higher autonomy (free from hyperparameter tuning), it is retained as the core mechanism in $E^2$LoRA.

## H   ROBUSTNESS ANALYSIS OF RMI-BASED THRESHOLDING

To empirically validate the robustness of our method against potential fluctuations in the Relative Mutual Information (RMI) matrix—and by extension, the derived thresholds $\tau$—we conducted a noise perturbation analysis on the image classification task.

### H.1   PERTURBATION METHODOLOGY

Since the adaptive thresholds are computed directly from the RMI statistics, introducing noise to the RMI matrix simulates scenarios where the layer-wise similarity estimation might contain errors or fluctuations. We injected Gaussian noise $\epsilon$ into the computed RMI matrix prior to the partitioning phase:

$$\tilde{\text{RMI}} = \text{RMI} + \epsilon, \quad \text{where } \epsilon \sim \mathcal{N}(0, \sigma^2), \tag{9}$$

where $\sigma$ controls the magnitude of the perturbation. We evaluated the downstream performance across a spectrum of noise levels, ranging from weak ($\sigma = 0.001$) to large ($\sigma = 0.05$).

## H.2 RESULTS AND DISCUSSION

The impact of RMI perturbation on model parameters and classification accuracy is summarized in Table 11.

Table 11: Impact of Gaussian noise perturbation on the RMI matrix. The experiment is conducted on image classification tasks. $\sigma$ denotes the standard deviation of the added noise.

| Noise Level ($\sigma$) | Intensity | Params | Accuracy (%) |
|---|---|---|---|
| 0.001 | Weak | 0.67 M | 89.81 |
| 0.005 | Weak | 0.65 M | 89.87 |
| 0.01 | Medium | 0.63 M | 89.85 |
| 0.05 | Large | 0.55 M | 89.30 |

**Stability under Minor Perturbations.** As shown in Table 11, under small to moderate perturbations ($\sigma \in [0.001, 0.01]$), the model performance remains highly stable, fluctuating by less than 0.1%. This indicates that our greedy grouping strategy is not brittle; it does not require "perfect" RMI estimation to function effectively and maintains high performance even with noisy structural signals.

**Significance of RMI Structure.** Conversely, when significantly larger noise ($\sigma = 0.05$) is introduced, disrupting the inherent similarity structure of the RMI matrix, we observe a noticeable degradation in performance (dropping to 89.30%). This performance drop confirms that the original RMI matrix captures essential structural information necessary for near-optimal partitioning. Randomizing or heavily distorting this structure prevents the algorithm from identifying the correct layer boundaries, thereby justifying the validity of our RMI-based design.

## I ROBUSTNESS TO SAMPLING VARIABILITY

To explicitly address the stability of our method with respect to the specific choice of data samples, we evaluated the consistency of the derived adaptive configurations—specifically, the interval partitioning and rank allocation—across different random seeds. This analysis ensures that our entropy-based metrics are robust to the randomness inherent in subset sampling.

### I.1 CONSISTENCY ANALYSIS

We re-computed all entropy-based quantities, including the Relative Mutual Information (RMI), the resulting interval boundaries, and the final adaptive rank allocation, using three distinct random seeds (0, 18, and 42). Each seed corresponds to an independently sampled mini-subset of data. We then trained the model using these configurations and recorded the final downstream performance.

Table 12 illustrates the results on the EuroSAT dataset for the $k\_proj$ module of the CLIP-ViT-B/16 model. We report the number of generated intervals, the specific rank assigned to each interval, and the final classification accuracy.

**Structural Stability.** As demonstrated in Table 12, the structural decisions made by our algorithm are remarkably consistent. The interval partitioning structure (represented by the number of layers in each interval: [2, 1, 3, 1, 3, 2]) and the corresponding rank allocations are identical across all three seeds. This indicates that the gradient-based entropy metrics capture intrinsic properties of the model layers that are invariant to the specific random subset of data used for estimation.

**Performance Stability.** Consequently, the final downstream performance exhibits minimal variance, fluctuating within a narrow margin (approximately $\pm 0.2\%$). We observed similar behavior across other modules (e.g., Attention and FFN blocks) and datasets. These findings confirm that $E^2$LoRA yields highly stable adaptive decisions that are robust not only to the size of the subset but also to the specific *choice* of samples.

Table 12: Robustness of interval partitioning and rank allocation across different random seeds on EuroSAT (CLIP-ViT-B/16, $k\_$proj module). The subset choice varies with the seed, yet the structural decisions remain identical.

| Seed | # Intervals | Rank Allocation (Interval ID: Rank) | Acc (%) |
|---|---|---|---|
| 0 | [2, 1, 3, 1, 3, 2] | {0: 9, 1: 9, 2: 7, 3: 8, 4: 7, 5: 8} | 98.56 |
| 18 | [2, 1, 3, 1, 3, 2] | {0: 9, 1: 9, 2: 7, 3: 8, 4: 7, 5: 8} | 98.28 |
| 42 | [2, 1, 3, 1, 3, 2] | {0: 9, 1: 9, 2: 7, 3: 8, 4: 7, 5: 8} | 98.52 |

## J EXTENDED PARAMETER-MATCHED COMPARISONS

In the main text, we demonstrated the superior parameter efficiency of $E^2$LoRA on the GSM-8K task. To rigorously validate the generalizability of these findings, we expanded the parameter-matched comparisons to cover Computer Vision (CV) and Natural Language Understanding (NLU) modalities. Specifically, we compared $E^2$LoRA against standard LoRA on the CV task (using CLIP-ViT-B/16) and the NLU task (using RoBERTa on GLUE), strictly controlling the trainable parameter counts.

### J.1 EXPERIMENTAL SETUP

We aligned the parameter budgets by adjusting the rank settings. Due to the efficient parameter-sharing mechanism in $E^2$LoRA, our method can utilize a higher base rank while maintaining a low parameter footprint. Consequently, we compared:

- **Low Budget:** Standard LoRA (Rank 4) vs. $E^2$LoRA (Base Rank 8).
- **Medium Budget:** Standard LoRA (Rank 8) vs. $E^2$LoRA (Base Rank 16).

### J.2 RESULTS AND ANALYSIS

The comparative results are summarized in Table 13. At identical parameter budgets, $E^2$LoRA consistently outperforms standard LoRA across both domains.

Table 13: Performance comparison between Vanilla LoRA and $E^2$LoRA under matched parameter counts. $E^2$LoRA achieves higher performance using the same or fewer parameters.

| Method | Config | CV (CLIP-ViT) | | NLU (RoBERTa) | |
|---|---|---|---|---|---|
| | | Params | Acc (%) | Params | Score |
| LoRA | Rank 4 | 0.66 M | 88.03 | 0.15 M | 85.20 |
| **$E^2$LoRA** | **Rank 8** | **0.66 M** | **89.81** | **0.16 M** | **85.57** |
| LoRA | Rank 8 | 1.31 M | 89.08 | 0.30 M | 85.87 |
| **$E^2$LoRA** | **Rank 16** | **1.33 M** | **90.45** | **0.32 M** | **86.22** |

**Consistent Superiority.** As shown in Table 13, $E^2$LoRA significantly surpasses the parameter-matched LoRA baselines. For instance, in the CV task with a budget of $\sim$0.66M parameters, $E^2$LoRA exceeds LoRA by +1.78%. Similarly, in the NLU task, $E^2$LoRA consistently achieves higher GLUE scores at both budget levels. This confirms that our framework's benefits—stemming from adaptive interval partitioning and rank allocation—are robust and meaningful even when strictly controlling for model size.

## K  ADDITIONAL EXPERIMENTS ON COMMONSENSE REASONING

To further verify the robustness of our method on diverse reasoning tasks beyond GLUE and Math benchmarks, we extended our evaluation to the CommonSenseQA benchmark using LLaMA-3.1-8B. This suite encompasses multiple challenging subtasks, including ARC-Challenge (ARC-C), ARC-Easy (ARC-E), BoolQ, OpenBookQA (OBQA), PIQA, and SIQA. These tasks evaluate the model's instruction-style reasoning capabilities without relying on long-context or code-specific adaptations.

The results, summarized in Table 14, demonstrate that $E^2$LoRA maintains a strong efficiency-performance balance even on complex commonsense reasoning tasks. Specifically, $E^2$LoRA achieves a slight overall improvement (+0.09 points in average accuracy) while utilizing only 48% of the trainable parameters required by standard LoRA (3.26M vs. 6.82M).

Notably, our method exhibits distinct performance characteristics across subtasks. It excels in tasks requiring nuanced reasoning, achieving significant gains on **BoolQ** (+5.11 points) and **SIQA** (+3.05 points). While there are minor performance dips in other areas (e.g., ARC-C), these are effectively offset by the gains in high-complexity tasks. This behavior aligns with our entropy-guided allocation strategy, which prioritizes parameter budget for high-information layers. These findings further reinforce the generality of $E^2$LoRA across different model families (LLaMA-3.1), scales, and task types.

Table 14: Performance comparison on CommonSenseQA using LLaMA-3.1-8B. $E^2$LoRA achieves comparable or better average performance with significantly fewer parameters.

| Method | #Params | Avg. | ARC-C | ARC-E | BoolQ | OBQA | PIQA | SIQA |
|---|---|---|---|---|---|---|---|---|
| LoRA | 6.82M | 82.18 | **82.50** | **93.09** | 64.95 | **86.40** | **88.46** | 77.67 |
| Ours ($E^2$LoRA) | 3.26M | **82.27** | 81.14 | 91.45 | **70.06** | 84.20 | 86.07 | **80.72** |

## L  EXPERIMENTS ON LARGER AND DIVERSE MODELS.

To further verify the robustness, scalability, and generality of our method, we conducted experiments on a diverse set of large language models (LLMs) using math benchmarks. Specifically, we evaluated $E^2$LoRA against the standard LoRA on **Llama-2-13B** (Dense), **Qwen3-8B** (Dense), and **GPT-OSS-20B** (a Mixture-of-Experts, MoE, model). As shown in Table 15, these models represent different scales, families, and architectures.

The results consistently demonstrate $E^2$LoRA's superior efficiency and performance. On Llama-2-13B, $E^2$LoRA achieves higher accuracy (+0.88) with approximately half the parameters of LoRA. On Qwen3-8B, it matches LoRA's accuracy using only 53% of the parameters. Most impressively, on the complex GPT-OSS-20B (MoE) model, $E^2$LoRA not only reduces parameters by 56% but also surpasses LoRA's accuracy by a significant margin of 1.14 points. These findings strongly validate that our dual-adaptive framework is a generalizable approach, effectively delivering a better performance-efficiency trade-off across various LLM designs. Further analysis, including layer-wise gradient similarity and entropy visualizations for these models, is provided in Appendix L, which confirms that the principles motivating our method are intrinsic properties of LLMs.

Table 15: Performance of fine-tuning diverse large models on math benchmarks. $E^2$LoRA consistently achieves better or equal performance with significantly fewer parameters across different model scales, families, and architectures (Dense vs. MoE).

| Model (Architecture) | Method | #Params | Accuracy (%) |
|---|---|---|---|
| Llama-2-13B (Dense) | LoRA | 13.11M | 52.94 |
| | Ours ($E^2$LoRA) | 6.55M | **53.82** |
| Qwen3-8B (Dense) | LoRA | 7.67M | 82.49 |
| | Ours ($E^2$LoRA) | 4.06M | **82.49** |
| GPT-OSS-20B (MoE) | LoRA | 3.98M | 82.03 |
| | Ours ($E^2$LoRA) | 1.75M | **83.17** |

## M    DETAILED ALGORITHM

The detailed processes of Local Similarity-based Sharing(LSS) are shown in Algorithm 2, the detailed processes of Heterogeneity-based Rank Allocation(HRA) are shown in Algorithm 3.

---

**Algorithm 2:** Local Similarity-based Sharing

---

**Notations:**
Let $R[i][j]$ denote RMI between $\mathbf{G}^{(i)}$ and $\mathbf{G}^{(j)}$, $\mathcal{H}$ be gradient entropies, $\mathcal{L} = [L_1, ..., L_N]$ be layer indices.

**Input:** Model gradients $\{\mathbf{G}^{(1)}, ..., \mathbf{G}^{(N)}\}$
Proxy entropies $\mathcal{H} = \{H_1, ..., H_N\}$
Number of layers $N$
**Output:** Sharing intervals $\mathcal{S} = \{[s_k, e_k]\}_{k=1}^K$
**for** $i \in \mathcal{L}$ **do**
  **for** $j \in \mathcal{L}$ **do**
    **if** $j \neq i$ **then**
      $R[i][j] \leftarrow$ ComputeRMI$(\mathbf{G}^{(i)}, \mathbf{G}^{(j)})$

$\mathcal{S} \leftarrow \emptyset$ // Initialize the interval set
$p \leftarrow 1$ // Pointer to current layer
**while** $p \leq N$ **do**
  $s \leftarrow p$ // Start index of interval
  $\tau \leftarrow \frac{1}{N-1} \sum_{k \neq s} R[s][k]$ // Threshold
  $e \leftarrow s$ // Initialize end layer
  **while** $e < N$ **and** $R[e+1][s] \geq \tau$ **do**
    $e \leftarrow e + 1$ // Expand interval
  $\mathcal{S} \leftarrow \mathcal{S} \cup \{[s, e]\}$ // Update the interval set
  $p \leftarrow e + 1$ // Move to next layer
**return** $\mathcal{S}$

---

**Algorithm 3:** Heterogeneity-based Rank Allocation

---

**Notations:**
Let $\mathcal{M} = \{\mathbf{W}^{(l)}\}_{l=1}^N$ be the model, $\mathcal{I} = \{[s_k, e_k]\}_{k=1}^K$ be sharing intervals, $r_{vanilla}$ be the base LoRA rank.

**Input:** Model $\mathcal{M}$ with $N$ layers
Base rank $r_{vanilla}$
Sharing intervals $\mathcal{I}$
**Output:** Allocated ranks $\mathcal{R} = \{r_k\}_{k=1}^K$
$\mathcal{R} \leftarrow \emptyset$ // Initialize the rank set
$R_{\text{total}} \leftarrow N \times r_{vanilla}$ // Total rank budget
$\mathcal{H} \leftarrow \emptyset$ // Interval entropy container
**for** *each interval* $[s_k, e_k] \in \mathcal{I}$ **do**
  $H_k \leftarrow \max_{s_k \leq l \leq e_k} H(\mathbf{G}^{(l)})$ // Max entropy
  $\mathcal{H} \leftarrow \mathcal{H} \cup \{H_k\}$
$H_{\text{sum}} \leftarrow \sum \mathcal{H}$ // Total heterogeneity measure
**for** *each interval* $[s_k, e_k] \in \mathcal{I}$ **do**
  $\alpha_k \leftarrow \mathcal{H}[k]/H_{\text{sum}}$ // Allocation factor
  $r_k \leftarrow$ round$(\alpha_k \times R_{\text{total}})$ // Rank allocation
  $\mathcal{R} \leftarrow \mathcal{R} \cup \{r_k\}$
**return** $\mathcal{R}$

---

## N    LIMITATIONS AND FUTURE WORK

Our work has demonstrated that E$^2$LoRA consistently achieves an excellent balance of efficiency and effectiveness across diverse tasks, modalities, and various sizes of pre-trained models, often

surpassing existing methods. Despite these promising results, several avenues remain for further exploration.

First, our current experimental validation does not include fine-tuning on significantly larger models, such as Llama3-70B.

Second, while $E^2LoRA$ focuses on LoRA, the proposed entropy-based analysis is orthogonal to the specific adaptation mechanism. The insights regarding local similarity and layer-wise information heterogeneity rely solely on pre-trained weights and gradients. Therefore, future work could explore applying the E²LoRA framework to other PEFT methods, such as adaptive budget allocation for Adapters or token-wise optimization in Prefix-tuning.

Third, while $E^2LoRA$ has been successfully validated across both natural language and vision modalities, its generalizability to a broader range of model types and tasks warrants further investigation. Future work will explore the application of $E^2LoRA$ to emerging domains such as visual language models and visual question answering.

## O  LLM USAGE DISCLOSURE

In the preparation of this paper, we used the large language model to assist with language refinement, including improving and checking potential grammatical issues as well as enhancing clarity and readability. The model was not used to generate scientific content, ideas, experiments, or analyses. The authors take full responsibility for the accuracy and integrity of the paper's content.

