# OpenReview forum: "E²LoRA: Efficient and Effective Low-Rank Adaptation with Entropy-Guided Adaptive Sharing"
_ICLR.cc/2026/Conference — ICLR 2026 Poster_

### Official Review · Reviewer_CKkS · 2025-10-25

**Soundness:** 2
**Presentation:** 3
**Contribution:** 2
**Rating:** 2
**Confidence:** 3

**Summary:**

LoRA and other PEFT methods are efficient, but pushing further with cross-layer sharing or lower ranks often hurts representational capacity and degrades performance. E2LoRA addresses this by analyzing gradients with a proxy-entropy lens and drawing two key insights: local similarity that adjacent layers tend to carry more similar information and are thus suitable for blockwise sharing; and layer-wise heterogeneity, different layers (or intervals) carry very different amounts of information and should receive differentiated capacity (rank). Building on this, E2LoRA proposes a dual-adaptive framework: LSS (Local Similarity Sharing) uses relative mutual information with adaptive thresholds to greedily partition adjacent, information-similar layers into shared intervals and restrict sharing within each interval; HRA (Heterogeneity-aware Rank Allocation) then uses the maximum proxy entropy within each interval to allocate ranks under a fixed overall budget, granting higher ranks to more informative intervals.

**Strengths:**

1. The “local similarity + layer-wise heterogeneity” perspective operationalized via gradient proxy-entropy and relative mutual information offers a clear mechanism and transfers across backbones and tasks.

2. The dual-adaptive scheme (partition + rank allocation) is nearly plug-and-play with existing LoRA/ShareLoRA pipelines, reducing trainable parameters under a fixed budget while largely preserving (or slightly improving) performance.

**Weaknesses:**

1. While the paper targets both efficiency and effectiveness, the reported gains over baselines are not striking. Many tables show modest uplifts at comparable or lower trainable-parameter budgets.

2. Several LoRA-family methods in 2025 already achieve very low trainable-parameter regimes (e.g., COLM-2025 LoRI). The latest baseline here appears to be DoRA (Feb 2024). Given the pace of PEFT, comparisons should include newer 2024–2025 lines (ICLR/ACL/ICML 2025) to establish contemporaneous relevance.

3. Related work coverage is incomplete for the entropy/gradient/shared-layer thread. The method is positioned as entropy/gradient-guided sharing, yet closely related recent works are missing or under-discussed (e.g., BSLoRA, ICML 2025; LoRA-GA, NeurIPS 2024; LoRA-One, ICML 2025). A more thorough placement of these lines is needed (the 3 papers I mentioned are just an example; the related work is too weak ).

4. RoBERTa-base and ViT-B are relatively small by today’s standards. Although Llama-3.1-8B is included, a stronger LLM sweep (more family/bases, more diverse instruction/reasoning/long-context/code tasks) is important to substantiate generality.

5. Stability/robustness of the entropy-RMI partitioning is under-analyzed.



In summary, the paper targets peft via entropy/gradient-guided layer sharing and adaptive rank allocation. The direction is reasonable, the current presentation feels under-polished: the motivation is not sharply articulated, the method reads incremental relative to recent PEFT progress, and the writing/positioning leaves key claims under-substantiated. The paper doesn’t provide a new conceptual lens on PEFT; it reads more as an engineering tweak than a step that changes how we understand parameter sharing.

**Questions:**

See weakness.

---

> ### Author Response · Authors · 2025-11-26
> **Response to Reviewer CKkS (1)**
>
> ```
> Weakness 1
> While the paper targets both efficiency and effectiveness, the reported gains over baselines are not striking. Many tables show modest uplifts at comparable or lower trainable-parameter budgets.
> ```
> **Response to weakness 1:**
>
> We appreciate the reviewer’s feedback on the reported gains, but we want to claim that the main value of E²LoRA lies not in a marginal increase in performance but in its ability to achieve **significant parameter reduction** while **preserving or even improving performance with adaptive** division of shared intervals and allocation of ranks.
>
> The main contribution of E²LoRA is not merely a small uplift in performance but rather the **ability to reduce trainable parameters by approximately 50%** without sacrificing performance. This is a **fundamentally different approach** compared to many other methods that focus only on either reducing parameters at the cost of performance or boosting performance without considering parameter efficiency.
>
> To highlight this, we have shown that **E²LoRA** consistently outperforms **VeRA** and **LoRA variants** under comparable or lower parameter budgets. Even when performance improvements are modest, **the ability to halve the parameter count** while maintaining or improving performance provides **significant practical benefits** in terms of memory usage and computation efficiency.
>
> |  | cv |  | nlu |  | nlg(math) |  |
> | --- | --- | --- | --- | --- | --- | --- |
> |  | params | result | params | result | params | result |
> | **dora-rank4** | **0.75M** | **88.15** | **0.17M** | **84.76** | **3.74M** | **67.40** |
> | dora-rank8 | 1.39M | 89.14 | 0.31M | 84.77 | 7.14M | 69.17 |
> | **adalora-rank4** | **0.66M** | **76.63** | **0.15M** | **83.94** | **3.41M** | **56.56** |
> | adalora-rank8 | 1.31M | 79.72 | 0.30M | 84.33 | 6.82M | 70.63 |
> | lora | 1.31M | 89.08 | 0.30M | 85.87 | 6.82M  | 70.21 |
> | **E2LoRA** | **0.66M** | **89.81** | **0.16M** | **85.57** | **3.66M** | **70.51** |
>
> We understand that the reported gains over baselines might appear modest, but we would like to emphasize the consistent efficiency improvements across **a variety** of tasks, models, and parameter budgets.
>
> - NLU (GLUE with RoBERTa-Base): E²LoRA reduces parameters by 50% compared to vanilla LoRA and maintains near-identical performance on average across all GLUE tasks.
> - CV (CLIP ViT-B/16 on 7 datasets): E²LoRA demonstrates significant performance gains **over LoRA** and **all other variants** including **VeRA** and **LoRA-FA** on various image classification benchmarks, where traditional methods either suffer accuracy degradation or yield only minimal improvements.
> - NLG (Math/Code tasks on LLaMA-3.1 and LLaMA-2-13B): The performance improvement in tasks like GSM8K and HumanEval is more significant when using E²LoRA, even with reduced parameters (50% of LoRA’s parameter count).
>
> To emphasize the effectiveness of our approach in terms of both **performance and efficiency**, we’ve conducted **parameter-controlled comparisons** where we match the trainable parameters of E²LoRA to those of LoRA and its variants. The results show that with almost the same parameters, variants with our framework consistently perform better than the variants themselves.
>
> Another **core advantage of E²LoRA is** **its generalizability across multiple architectures and tasks**. When combined with **LoRA, ShareLoRA, DoRA**(new added)**,** and **VeRA**(new added)**,** the method **still achieves a substantial parameter reduction** (50%) without compromising performance.
>
> | Model | Dataset | Method | Params | Accuracy/Score |
> | --- | --- | --- | --- | --- |
> | Roberta | GLUE | DoRA | 0.31M | 84.77 |
> |  |  | E2DoRA | 0.15M | 84.79 |
> |  |  | VeRA | 0.04M | 83.83 |
> |  |  | E2VeRA | 0.013M | 84.31 |
> | Clip-ViT | CV tasks | DoRA | 1.39M | 89.14 |
> |  |  | E2DoRA | 0.75M | 89.82 |
> |  |  | VeRA | 0.10M | 88.97 |
> |  |  | E2VeRA | 0.05M | 89.13 |
> | llama3.1-8B | GSM8K | DoRA | 7.14M | 69.17 |
> |  |  | E2DoRA | 2.47M | 71.03 |

---

> ### Author Response · Authors · 2025-11-26
> **Response to Reviewer CKkS (2)**
>
> ```
> Weakness 2
> Several LoRA-family methods in 2025 already achieve very low trainable-parameter regimes (e.g., COLM-2025 LoRI). The latest baseline here appears to be DoRA (Feb 2024). Given the pace of PEFT, comparisons should include newer 2024–2025 lines (ICLR/ACL/ICML 2025) to establish contemporaneous relevance.
> ```
> **Response to weakness 2:**
>
> We thank the reviewer for pointing out the rapid developments in the PEFT field. We implement **LoRI** (COLM-2025) and conducted extensive comparisons on CV and GLUE tasks.
>
> |  | cv |  | nlu |  |
> | --- | --- | --- | --- | --- |
> |  | params | result | params | result |
> | VeRA | 0.10M | 88.97 | 0.04M | 83.83 |
> | E2VeRA | 0.05M | 89.13 | 0.013M | 84.31 |
> | LoRI-S | 0.066M | 89.47 | 0.015M | 78.26 |
> | E2LoRI-S | 0.033M | 89.43 | 0.007M(7K) | 78.96 |
>
> As shown, even against a cutting-edge baseline like LoRI-S—which achieves impressively low parameter regimes—our entropy-guided adaptive sharing and rank allocation (LSS + HRA) yield further **parameter reductions** (up to ~50%) while maintaining or improving performance. On NLU, E²LoRI-S boosts accuracy by 0.7 points with half the parameters, underscoring that **E²LoRA captures intrinsic layer-wise information heterogeneity that LoRI's dynamic masking may overlook**. This positions E²LoRA not merely as a competitor, but as a **universal enhancer** for emerging PEFT methods.
>
> Beyond performance, a key efficiency distinction emerges when comparing against both LoRI and VeRA:
>
> - **LoRI** relies on complex dynamic adjustments and sparse updates, incurring **higher training latency and memory overhead**. For instance, training the sparse LoRI-S variant requires first completing a full LoRI-D process to derive the necessary masks, which—despite slashing parameters—introduces substantial time costs (often 1.5–2x longer than baseline LoRA).
> - **VeRA**, while elegantly sharing low-rank matrices to minimize parameters, employs a **fixed, global sharing pattern** that can lead to suboptimal adaptation across heterogeneous layers, sometimes necessitating hyperparameter retuning per task and adding minor initialization overhead.
>
> In contrast, **E²LoRA** is a lightweight, **one-shot pre-computation** approach: gradient-based entropy and similarity metrics are computed in ~30 seconds upfront, after which training mirrors vanilla LoRA with **zero additional computational cost**. This plug-and-play nature allows seamless integration with LoRI or VeRA, amplifying their efficiency without their drawbacks.
>
> These enhancements confirm E²LoRA's contemporaneous value, offering a principled, low-overhead path to superior parameter-effectiveness trade-offs in the evolving PEFT landscape.
>
> ---
>
> ```
> Weakness 3
> Related work coverage is incomplete for the entropy/gradient/shared-layer thread.
> ```
> **Response to weakness 3:**
>
> We are grateful to the reviewer for identifying these articles. In the revised manuscript, we have added some updated articles to the "Related Work" section, including the three articles you provided, as well as ALoRA, DyLoRA, IncreLoRA, RA-LoRA, and HydraLoRA.
>
> Below, we detail the critical distinctions and comparisons included in the revision:
>
> Comparison with BSLoRA (ICML 2025): **BSLoRA** proposes a **fixed** hierarchical sharing structure: it decomposes weights into *Local* (specific), *Intra-layer* (shared within a layer), and *Inter-layer* (shared across **all** layers) components. **While E²LoRA** employs **data-driven, adaptive topology**: We calculate the *Relative Mutual Information (RMI)* between layers to dynamically group them into **contiguous blocks**.
>
> Comparison with LoRA-GA (NeurIPS 2024) & LoRA-One (ICML 2025): Both **LoRA-GA** and **LoRA-One** utilize gradient information, but they focus on **Initialization** and **Optimization Dynamics**, whereas E²LoRA focuses on trade-off between efficiency and effectiveness. They use gradients to determine the initialization of A&B. While E²LoRA uses gradient to determine **what the rank of A/B should be, and which layers should share them.**
>
> Additionally, we incorporate discussions on other relevant works in **rank allocation and parameter sharing:** Rank Allocation Technique: ALoRA [1], IncreLoRA [2], RA-LoRA [3], DyLoRA[4]. Parameter Sharing: HydraLoRA [5]. For a deeper discussion on these methods, please refer to our **Response to W1 of Reviewer VLBt (1).**
>
> [1] ALoRA: Allocating Low-Rank Adaptation for Fine-tuning Large Language Models
> [2] IncreLoRA: Incremental Parameter Allocation Method for Parameter-Efficient Fine-tuning
> [3] RA-LoRA: Rank-Adaptive Parameter-Efficient Fine-Tuning for Accurate 2-bit Quantized Large Language Models
> [4] DyLoRA: Parameter-Efficient Tuning of Pretrained Models using Dynamic Search-Free Low Rank Adaptation
> [5] HydraLoRA: An Asymmetric LoRA Architecture for Efficient Fine-Tuning

---

> ### Author Response · Authors · 2025-11-26
> **Response to Reviewer CKkS (3)**
>
> ```
> Weakness 4
> A stronger LLM sweep (more family/bases, more diverse instruction/reasoning/long-context/code tasks) is important to substantiate generality.
> ```
> **Response to weakness 4:**
>
> Thank you for mentioning the importance of evaluating our method on more LLMs and tasks. In response to this, we have **significantly expanded our experiments** during the rebuttal period to include **different model families** (beyond LLaMA), **larger scales** (up to 20B), different architectures(MoE-based), and **more diverse reasoning tasks**.
>
> We extended our evaluation to include: **Qwen3-8B:** To test generality on a different architecture widely used in the community. **GPT-OSS-20B:** To validate the method on larger-scale models beyond the 7B/8B regime, and it is a MoE-based LLM.
>
> The results on these new models and tasks follow the exact same pattern as our previous LLaMA experiments: **E²LoRA achieves comparable or better performance with ~50% fewer trainable parameters.**
>
> | Model | Method | Params | Accuracy (%) |
> | --- | --- | --- | --- |
> | Qwen3-8B (Dense) | LoRA | 7.67M | 82.49 |
> |  | E2LORA | 4.06M | 82.49 |
> | GPT-OSS-20B (MoE) | LoRA | 3.98M | 82.03 |
> |  | E2LORA | 1.75M | 83.17 |
>
> To further address the need for diverse **reasoning tasks**, we also fine-tuned **LLaMA-3.1-8B** on **CommonSenseQA** (a challenging commonsense reasoning benchmark comprising multiple subtasks like ARC-C, ARC-E, BoolQ, OpenBookQA, PIQA, and SIQA). This evaluates E²LoRA's robustness on instruction-style reasoning without long-context or code-specific adaptations, complementing our existing Math/Code sweeps.
>
> The results (added to Appendix F) show E²LoRA maintaining a strong efficiency-performance trade-off:
>
> | CommonSense | Params | Performance | arc-c | arc-e  | boolq | openbookqa     | piqa | siqa           |
> | --- | --- | --- | --- | --- | --- | --- | --- | --- |
> | LoRA | 6.82M | 82.18 | 82.50 | 93.09 | 64.95 | 86.40 | 88.46 | 77.67 |
> | E2LoRA | 3.26M | 82.27 | 81.14 | 91.45 | 70.06 | 84.20 | 86.07 | 80.72 |
>
> E²LoRA achieves a **slight overall improvement** (+0.09 points) with only **48%** of LoRA's parameters. It excels on subtasks requiring nuanced reasoning (e.g., +5.11 points on BoolQ, +3.05 on SIQA), while minor dips on others (e.g., ARC-C) are offset by the entropy-guided allocation prioritizing high-information layers. This confirms E²LoRA's adaptability to reasoning-heavy scenarios, reinforcing its generality across model families, scales, architectures, and task types.
>
> ```
> Weakness 5
> Stability/robustness of the entropy-RMI partitioning is under-analyzed.
> ```
> **Response to weakness 5:**
>
> We thank the reviewer for this crucial question. We have conducted additional analyses to demonstrate that the **entropy-RMI partitioning is both robust (stable against perturbations) and effective (near-optimal).**
>
> As discussed in **Section 5 (Discussion)** and shown in **Figure 5** of the original paper, E²LoRA stabilizes rapidly. We test 8-512 steps, the entropy profile and resulting performance keeps relatively stable.
>
> To rigorously stress-test the stability of the partitioning algorithm itself, we conducted a **noise injection experiment** during the rebuttal.
>
> Since the thresholds ($\tau$) derive directly from the RMI matrix, adding noise to the RMI matrix simulates errors or fluctuations in the thresholding logic. We added Gaussian noise $$\epsilon \sim \mathcal{N}(0, \sigma^2_{noise})$$ to the computed RMI matrix before partitioning and measured the impact on performance. Results on image classification datasets are shown below.
>
> **Results:**
>
> | noise | Params | Acc |
> | --- | --- | --- |
> | 0.001(weak) | 0.67M | 89.81 |
> | 0.005(weak) | 0.65M | 89.87 |
> | 0.01(medium) | 0.63M | 89.85 |
> | 0.05(large) | 0.55M | 89.30 |
>
> **Robustness (Small Noise):** Under small to moderate perturbations ($\sigma=0.001 \sim 0.005$), the performance fluctuates negligibly. This proves that the method is robust; it does not require "perfect" thresholds to work effectively.
>
> **Optimality (Large Noise):** When significantly larger noise ($\sigma=0.05$) disrupts the RMI structure, performance degrades. This confirms that our computed RMI matrix and the derived thresholds indeed capture a **near-optimal partitioning** structure, and randomizing this structure harms the model.

---

> ### Author Response · Authors · 2025-11-26
> **Response to Reviewer CKkS (4)**
>
> **Response to final concern:**
>
> Thank you for your compliment and recognition on "The direction is reasonable". We respectfully argue that E²LoRA is **not merely an engineering tweak**, but a **principled framework** that introduces an **information-theoretic lens** to resolve the long-standing tension between parameter efficiency and model effectiveness. We address the concerns on motivation, novelty, and substantiation below.
>
> 1. **On "Motivation is not sharply articulated" and "Reads more as an engineering tweak"**
>
> We respectfully clarify that our motivation is grounded in a systematic and principled empirical analysis, rather than ad-hoc engineering adjustments.
>
> - **Principled Foundation:** Our work is the first to introduce an **information-theoretic perspective (entropy)** to the study of LoRA parameter sharing and rank allocation. Through *gradient-based proxy entropy*, we uncovered two critical, previously overlooked properties in pre-trained models: **"Local Similarity"** and **"Layer-wise Information Heterogeneity."**
> - **Theoretical Insight:** These findings provide principled motivation for our design. They explain *why* naive sharing methods (like global sharing) fail and theoretically guide *how* to design superior sharing strategies.
> - **Corroboration from other Reviewers:** We are encouraged that other reviewers recognize this conceptual contribution. Reviewer LoWT noted our analysis provides a *"potentially valuable and principled motivation... moving beyond naive global or fixed-block sharing,"* calling the perspective *"somewhat original."* Similarly, Reviewer wUSM highlighted that our work *"Introduces a novel entropy-based view of LoRA sharing and rank allocation."* These comments affirm that our approach represents a conceptually new direction, not merely a tweak.
>
> 2. **On "Method reads incremental relative to recent PEFT progress"**
>
> We argue that E2LORA addresses a fundamental contradiction that existing methods have failed to resolve:
>
> - **The Current Landscape:** Existing works typically prioritize either effectiveness (e.g., AdaLoRA, DoRA) at the cost of parameter efficiency, or efficiency (e.g., ShareLoRA, VeRA) often resulting in performance degradation.
> - **Our Unique Contribution:** E2LORA introduces a **dual-adaptive framework**  derived directly from our two novel insights. This allows us to significantly enhance efficiency while maintaining—or surpassing—original performance.
> - **Non-Incremental Value:** Furthermore, our framework is **orthogonal** to existing methods. As demonstrated in our additional experiments, it can be combined with performance-enhancing methods (like DoRA variants) or efficiency-focused methods (like VeRA variants). This proves our method provides a distinct, refined mechanism for parameter utilization rather than a simple incremental step.
>
> 3. **On "Writing/positioning leaves key claims under-substantiated"**
>
> We have strengthened our manuscript to ensure our key claims are robustly substantiated by empirical evidence:
>
> - **Claim: Universality of Insights.** Our observations of *Local Similarity* and *Information Heterogeneity* are consistent across diverse modalities. As shown in **Fig 1** and **Appendix B.1 (Fig 6, left)**, this holds for NLU, NLG, and Vision. We have further validated this on **Qwen and GPT-OSS (MoE-based) models** , confirming these are fundamental properties of pre-trained models.
> - **Claim: Excellent Efficiency-Effectiveness Balance.** **Tables 1, 2, and 3** demonstrate that E2LORA matches or exceeds baseline performance using approximately **50% of the parameters**. We have further verified this by applying our framework to more pre-trained weights and combining it with other methods (DoRA/VeRA), fully validating our core goal: significantly enhancing efficiency without compromising effectiveness.
> - **Claim: Superior Parameter Utilization.** **Table 5** proves that under an identical parameter budget, E2LORA consistently outperforms standard LoRA. We have extended this experiment to both CV and NLU tasks, confirming the conclusion broadly.
> - **Claim: The Entropy-Guided Mechanism is Crucial.** Our ablation studies (**Figure 3 and Table 4**) demonstrate the necessity of our entropy-guided mechanisms. They outperform other similarity metrics, providing strong secondary evidence for the correctness of our insights and justifying the use of entropy to guide layer-wise parameter sharing.
>
> In conclusion, we respectfully argue that our work presents a **novel, principled, and empirically validated approach** to PEFT. By introducing a novel lens, we provide new insights into the structural properties of pre-trained models. The resulting dual-adaptive method (E2LORA) effectively resolves the long-standing tension between efficiency and effectiveness in LoRA sharing methods. We believe this is a conceptual contribution that changes how we approach parameter sharing, rather than just an engineering tweak.

---

### Official Review · Reviewer_LoWT · 2025-10-27

**Soundness:** 2
**Presentation:** 2
**Contribution:** 2
**Rating:** 2
**Confidence:** 4

**Summary:**

This paper proposes E²LoRA, a parameter-efficient fine-tuning method based on LoRA. It first analyzes pre-trained models using a gradient-based proxy entropy, identifying "Local Similarity" (adjacent layers share information) and "Layer-wise Information Heterogeneity" (layers contain different amounts of information). Based on these insights, E²LoRA introduces a dual-adaptive framework involving: 1) Adaptive Sharing Interval Partitioning, grouping similar adjacent layers to share LoRA adapters based on inter-layer entropy similarity , and 2) Adaptive Rank Allocation, assigning LoRA ranks to these shared intervals based on the absolute proxy entropy of the layers within them. The goal is to significantly reduce trainable parameters (claiming approx. 50% reduction) while maintaining or improving performance compared to baseline LoRA and naive sharing methods. Experiments are conducted across NLU, NLG, and image classification tasks.

**Strengths:**

1. The proposed dual-adaptive mechanism, which adjusts both the sharing scope (intervals) and the capacity (rank) based on properties derived from the model itself, is conceptually appealing.


2. The paper generally presents its motivation and method clearly. Figure 2 provides a good overview of the framework.

3. The initial analysis using gradient-based proxy entropy to identify Local Similarity and Layer-wise Information Heterogeneity  provides a potentially valuable and principled motivation for designing adaptive parameter sharing strategies, moving beyond naive global or fixed-block sharing. This perspective is somewhat original.

**Weaknesses:**

1. Complexity vs. Benefit: The proposed E²LoRA framework adds significant complexity compared to standard LoRA or simple sharing schemes. It requires an initial phase of gradient computation on a data subset , proxy entropy calculation , Relative Mutual Information (RMI) matrix computation , adaptive thresholding and partitioning , and finally rank allocation based on interval entropy. While the paper claims parameter reduction, the reported performance gains over baselines are marginal (sometimes slightly lower or equivalent on average). This raises questions about whether the added complexity is justified by the benefits

2. Proxy Entropy Robustness: The method heavily relies on gradient-based proxy entropy calculated on a "mini-subset of the training dataset". The stability and reliability of this proxy measure are questionable. How sensitive is it to the choice and size of this subset? Does this entropy accurately capture the necessary information across different tasks, modalities, and model states? The paper lacks analysis on the robustness of this core component. Basing adaptive decisions on potentially unstable proxies undermines the method's soundness.

3. Insufficient Comparisons: The comparisons, especially regarding efficiency, could be stronger. For instance, while E²LoRA uses fewer parameters than standard LoRA (e.g., 0.16M vs 0.30M in Table 1 ), the performance difference isn't compelling. A more rigorous comparison would involve evaluating standard LoRA with a reduced rank to match E²LoRA's parameter count (or vice-versa, as done briefly in Table 5  but only for one model/task) across all experiments to isolate the benefit of the adaptive strategy itself versus just using fewer parameters. Comparisons against methods like VeRA, which achieve even lower parameter counts, also show mixed results, with E²LoRA sometimes underperforming despite using more parameters (though comparisons here are complicated by different base ranks).

4. Hyperparameter Sensitivity: The adaptive partitioning uses layer-specific thresholds $\tau_m$ derived from average RMI 20, and the rank allocation depends on a base rank $r_0$ 21and the total budget calculation22. The sensitivity to these aspects (especially $r_0$ and the implicit assumptions in the thresholding) is not adequately explored.

**Questions:**

1. Could the authors provide details on the computational overhead (time and memory) introduced by the initial gradient computation, entropy/RMI calculation, and partitioning/allocation steps? How does this compare to the overall fine-tuning time?

2. How robust is the proxy entropy measure and the resulting partitioning/ranking to the choice and size of the initial data subset used for gradient calculation? Have the authors experimented with different subsets?

3. Could the authors provide more extensive comparisons where baseline LoRA (and potentially other methods like AdaLoRA, DoRA) are configured to have the exact same number of trainable parameters as E²LoRA across the main benchmark tables (Tables 1, 2, 3)? This would allow for a fairer assessment of the proposed adaptive strategy's effectiveness versus simply reducing rank.

4. How sensitive is the adaptive partitioning algorithm (Algorithm 2 ) to noise or minor variations in the RMI matrix? Does the greedy approach guarantee a near-optimal partitioning?

---

> ### Author Response · Authors · 2025-11-26
> **Response to Reviewer LoWT (1)**
>
> ```
> Weakness 1:
> Complexity vs. Benefit: The proposed E²LoRA framework adds significant complexity compared to standard LoRA or simple sharing schemes. …… While the paper claims parameter reduction, the reported performance gains over baselines are marginal. …… This raises questions about whether the added complexity is justified by the benefits.
> Question 1:
> Could the authors provide details on the computational overhead ……? How does this compare to the overall fine-tuning time?
> ```
> **Response to weakness 1 & question 1:**
>
> 1. **Complexity Analysis.**
>
> We thank the reviewer for this key concern. While E²LoRA conceptually includes multiple steps (gradient collection, proxy entropy, RMI computation, interval partitioning, and rank allocation), these are **computationally lightweight** with **negligible overhead** in practice.
>
> **(a) All computations operate on frozen pre-trained weights**
>
> No backpropagation through adapters or optimization is required, only a **single-pass gradient collection** over a small data subset.
>
> **(b) Linear Complexity:** All steps scale as **O(L)** (model depth): entropy/RMI (local-window sparse), partitioning, and allocation.
>
> **(c) Actual measured overhead is extremely small**
>
> As reported in **Appendix D**, the entire additional procedure requires only **~30 seconds** for LLaMA-3.1-8B on a single A100 GPU — less than **2%** of total finetuning time(40+ mins).
>
> **(d) Precomputed results are fully reusable**
>
> Most importantly, entropy profiles and intervals are reusable across LoRA variants, runs, and hyper-parameters (fixed pre-trained weights/task), yielding **zero cost** for these different setups. We'll make it more clear in the revision.
>
> 2. **Is the Additional Complexity Worth It?**
>
> E²LoRA's goal sets it apart: unlike parameter-focused variants (e.g., LoRA-FA/VeRA, accepting severe accuracy drops) or accuracy-only methods (no efficiency guarantees), we **automatically reduce parameters ~50% while preserving/enhancing performance** across LoRA variants.
>
> This delivers superior generalization across NLU, CV, and NLG: cuts LoRA params **by half**; matches/surpasses vanilla LoRA; outperforms compression baselines; avoids aggressive sharing's losses. This is a **meaningful practical benefit**, especially under memory-limited training environments.
>
> A major advantage of the E²LoRA framework is **full orthogonality**: It can wrap performance-oriented variants like DoRA, or further compress efficiency-oriented variants like ShareLoRA and VeRA.
>
> Unlike static/hand-crafted designs in VeRA, LoRA-FA, and ShareLoRA (fixed heuristics like global freezing/uniform blocks, limiting task adaptation), **E²LoRA's data-driven sharing autonomously selects intervals/ranks** via entropy-guided gradients. This yields **superior generalization** across tasks: robust on saturated NLU, complex reasoning (GSM8K), and different modality (CV).
>
> Thus, we believe that investing a mere **30 seconds of one-time overhead** to unlock **~50% parameter savings** with **negligible performance loss** across such a wide range of tasks is, without doubt, a highly worthwhile trade-off.
>
> We have also experimentally demonstrated this by combining E²LoRA with **ShareLoRA,** **DoRA**, and **VeRA** both of which obtain further improvements. The relevant results of ShareLora have been presented in the paper, and only the results combined with Dora and Vera are shown here.
>
> | Model | Dataset | Method | Params | Accuracy/Score |
> | --- | --- | --- | --- | --- |
> | Roberta | GLUE | DoRA | 0.31M | 84.77 |
> |  |  | E2DoRA | 0.15M | 84.79 |
> |  |  | VeRA | 0.04M | 83.83 |
> |  |  | E2VeRA | 0.013M | 84.31 |
> | Clip-ViT | CV tasks | DoRA | 1.39M | 89.14 |
> |  |  | E2DoRA | 0.75M | 89.82 |
> |  |  | VeRA | 0.10M | 88.97 |
> |  |  | E2VeRA | 0.05M | 89.13 |
> | llama3.1-8B | GSM8K | DoRA | 7.14M | 69.17 |
> |  |  | E2DoRA | 2.47M | 71.03 |
>
> We therefore believe the additional initialization step is a **highly cost-effective and practically justified investment**, and we will clarify this more explicitly in the revised version.

---

> ### Author Response · Authors · 2025-11-26
> **Response to Reviewer LoWT (2)**
>
> ```
> Weakness 2:
> Proxy Entropy Robustness: The method heavily relies on gradient-based proxy entropy calculated on a "mini-subset of the training dataset". The stability and reliability of this proxy measure are questionable. How sensitive is it to the choice and size of this subset? Does this entropy accurately capture the necessary information across different tasks, modalities, and model states? The paper lacks analysis on the robustness of this core component. Basing adaptive decisions on potentially unstable proxies undermines the method's soundness.
> Question2:
> How robust is the proxy entropy measure and the resulting partitioning/ranking to the choice and size of the initial data subset used for gradient calculation? Have the authors experimented with different subsets?
> ```
> **Response to weakness2 & question2:**
>
> 1. Relationship Between Subset Size and Stability
>
> We appreciate the reviewer’s concern.
>
> We clarify that the “subset size” used for proxy entropy is fully determined by:
>
> $\text{Subset Size} = \text{Step} \times \text{Batch Size}.$
>
> This means that the **“Effect of Step”** experiment in the Discussion section (Sec. 5) and **Figure 5 (right)** is exactly an ablation of subset size. The performance curve keeps **stable across different steps regardless of GSM-8K and HumanEval**. This directly demonstrates that the proxy entropy is **not sensitive** to the size of the mini-subset and becomes stable extremely quickly.
>
> 2. Robustness to Subset Choice
>
> To explicitly address stability with respect to *which* samples are chosen, we re-compute entropy-based quantities: RMI, interval partitioning, and adaptive rank allocation, using three different random seeds (i.e., three independently sampled mini-subsets). For illustration, Table below reports the results on the EuroSAT dataset (CV) for the $k\_\text{proj}$ module of CLIP-ViT-B/16:
>
> | Seed | #Intervals | Rank Allocation{interval_id:rank} | Results |
> | --- | --- | --- | --- |
> | 0 | [2, 1, 3, 1, 3, 2] | {0: 9, 1: 9, 2: 7, 3: 8, 4: 7, 5: 8} | 98.56 |
> | 18 | [2, 1, 3, 1, 3, 2] | {0: 9, 1: 9, 2: 7, 3: 8, 4: 7, 5: 8} | 98.28 |
> | 42 | [2, 1, 3, 1, 3, 2] | {0: 9, 1: 9, 2: 7, 3: 8, 4: 7, 5: 8} | 98.52 |
>
> As shown, both the interval structure and the final rank allocation are identical across seeds, while the final performance varies only slightly within a narrow band (around $\pm 0.2$). We observe the same behavior on other attention/FFN modules and on other datasets: the intervals and rank allocations are either identical or only differ marginally across seeds, and the resulting performance fluctuates only within a very small range. This directly indicates that the proxy entropy yields highly stable adaptive decisions with respect to subset choice, not only subset size.
>
> 3. Cross-Modal and Cross-Task Robustness
>
> We have shown consistent entropy and similarity patterns across:
>
> - **Roberta-base (Encoder-only, NLP)**
> - **CLIP-ViT-B/16 (Vision Transformer)**
> - **LLaMA-3.1-8B (Decoder-only LLM)**
>
> Figures 1 and 6 in the paper show the Local Similarity and Layer-wise Heterogeneity are highly consistent across *very different* architectures.
>
> To further strengthen robustness, we conducted extensive entropy/RMI analysis on more **additional LLMs**: **Qwen3-8B, GPT-OSS-20B(MoE Architecture).** For each model, we visualize the RMI matrix, and **include them in the Appendix**. The same phenomena (local similarity and Layer-wise Information Heterogeneity) consistently appear across different architectures (dense vs. MoE), different scales (7B → 20B), different modalities (vision & NLP). Thus, the insights are not only stable, but universal across model families.

---

> ### Author Response · Authors · 2025-11-26
> **Response to Reviewer LoWT (3)**
>
> ```
> Weakness 3:
> Insufficient Comparisons: The comparisons, especially regarding efficiency, could be stronger. For instance, while E²LoRA uses fewer parameters than standard LoRA (e.g., 0.16M vs 0.30M in Table 1 ), the performance difference isn't compelling. A more rigorous comparison would involve evaluating standard LoRA with a reduced rank to match E²LoRA's parameter count (or vice-versa, as done briefly in Table 5 but only for one model/task) across all experiments to isolate the benefit of the adaptive strategy itself versus just using fewer parameters. Comparisons against methods like VeRA, which achieve even lower parameter counts, also show mixed results, with E²LoRA sometimes underperforming despite using more parameters (though comparisons here are complicated by different base ranks).
>
> Question 3:
> Could the authors provide more extensive comparisons where baseline LoRA (and potentially other methods like AdaLoRA, DoRA) are configured to have the **exact same number of trainable parameters as E²LoRA across the main benchmark tables** (Tables 1, 2, 3)? This would allow for a fairer assessment of the proposed adaptive strategy's effectiveness versus simply reducing rank.
> ```
> **Response to weakness 3 & question 3:**
>
> We appreciate the reviewer’s suggestion and **have expanded the parameter-matched comparisons across all modalities and tasks**, going far beyond the single case shown in Table 5 of the main submission.
>
> 1. Parameter-Matched Comparisons Against LoRA Across All Settings
>
> We follow the reviewer’s recommendation by **matching the trainable parameter count** between: Vanilla LoRA and E²LoRA, and repeating the experiments on CV (ViT-B/16) and NLU (RoBERTa). **At identical parameter budgets, E²LoRA significantly outperforms standard LoRA,** demonstrating that our E2LoRA framework including **adaptive interval partitioning and rank allocation,** offers meaningful benefits when parameters are controlled.
>
> |  |  | cv |  | nlu |  |
> | --- | --- | --- | --- | --- | --- |
> |  |  | params | result | params | result |
> | lora | rank4 | 0.66M | 88.03 | 0.15M | 85.20 |
> |  | rank8 | 1.31M | 89.08 | 0.30M | 85.87 |
> | E2LoRA | rank8 | 0.66M | 89.81 | 0.16M | 85.57 |
> |  | rank16 | 1.33M | 90.45 | 0.32M | 86.22 |
>
> 2. Comparisons With Performance Enhancing Variants
>
> To further isolate the benefit of the adaptive strategy, we conducted **parameter-matched comparisons** between: **LoRA, DoRA, AdaLoRA, E²LoRA.** Even when using the *simplest* varient (plain LoRA), E²LoRA outperforms DoRA and AdaLoRA under the same parameter budget. This result appears on **CV, GLUE, and MATH**, and highlights the effectiveness of E²LoRA.
>
> |  | cv |  | nlu |  | nlg(math) |  |
> | --- | --- | --- | --- | --- | --- | --- |
> |  | params | result | params | result | params | result |
> | **dora-rank4** | **0.75M** | **88.15** | **0.17M** | **84.76** | **3.74M** | **67.40** |
> | dora-rank8 | 1.39M | 89.14 | 0.31M | 84.77 | 7.14M | 69.17 |
> | **adalora-rank4** | **0.66M** | **76.63** | **0.15M** | **83.94** | **3.41M** | **56.56** |
> | adalora-rank8 | 1.31M | 79.72 | 0.30M | 84.33 | 6.82M | 70.63 |
> | lora | 1.31M | 89.08 | 0.30M | 85.87 | 6.82M  | 70.21 |
> | **E2LoRA** | **0.66M** | **89.81** | **0.16M** | **85.57** | **3.66M** | **70.51** |
>
> 3. VeRA Discussion
>
> We clarify the design goals: **VeRA** is an extreme compression method (very small param count) at the cost of **unnegligible accuracy degradation (e.g., VeRA performs 2.0 lower than LoRA on GLUE, and 2.58 lower on GSM-8k. )**. While **E²LoRA** aims to **maintain LoRA-level performance while reducing  parameters**.
>
> To directly address the reviewer’s concern, we also **combine our framework with VeRA**, applying our interval sharing and rank allocation to its learnable vectors. The experiment shows the same results that E²VeRA still achieves almost unchanged performance with nearly half the number of parameters compared with VeRA.
>
> |  | cv |  | nlu |  |
> | --- | --- | --- | --- | --- |
> |  | params | result | params | result |
> | vera | 0.10M | 88.97 | 0.04M | 83.83 |
> | E2VeRA | 0.05M | 89.13 | 0.013M | 84.31 |
>
> We would like to re-emphasize that **E²LoRA is not a new LoRA structure but a resource allocation framework that is orthogonal to almost ALL LoRA variants.** It applies before the adapter is inserted and can be paired with: LoRA, ShareLoRA, DoRA, VeRA, which are we have verified with experiments. Therefore, we believe many **LoRA variant can immediately obtain ~50% parameter savings without losing its own performance characteristics**, with only a ~30s one-time preprocessing cost. This increases the practicality of E²LoRA framework in real applications.

---

> ### Author Response · Authors · 2025-11-26
> **Response to Reviewer LoWT (4)**
>
> ```
> Weakness 4:
> Hyperparameter Sensitivity: The adaptive partitioning uses layer-specific thresholds derived from average RMI, and the rank allocation depends on a base rank and the total budget calculation. The sensitivity to these aspects (especially and the implicit assumptions in the thresholding) is not adequately explored.
> Question 4:
> How sensitive is the adaptive partitioning algorithm (Algorithm 2) to noise or minor variations in the RMI matrix? Does the greedy approach guarantee a near-optimal partitioning?
> ```
> **Response to weakness 4 & question 4:**
>
> We thank the reviewer for this thoughtful question. We are happy to clarify that **E²LoRA is designed to be largely self-adaptive**, requiring minimal manual tuning.
>
> We address the sensitivity regarding (1) the user-defined hyperparameters (Rank & Steps) and (2) the implicit internal thresholds (RMI-based partitioning) below.
>
> 1. **Clarification: Only Two User-Defined Hyperparameters**
>
> Contrary to the concern about multiple implicit assumptions, our framework effectively relies on only **two** standard hyperparameters, both of which are robust:
>
> - **Base Rank ($r_{\text{vanilla}}$):** This determines the total parameter budget ($Budget = L \times r_{\text{vanilla}}$). It is not a "new" hyperparameter but simply the target budget user sets for any LoRA method.
> - **Steps ($N$):** The number of steps for gradient collection.
>
> Crucially, the thresholds ($\tau$) are **NOT** hyperparameters. They are computed **automatically** via Eq. 4 ($\tau = \mu + \sigma$) based on the statistics of the RMI matrix. The user does *not* set or tune these thresholds; they adapt dynamically to the model's entropy profile.
>
> 2. **Robustness to User Hyperparameters (Rank & Step)**
>
> As discussed in **Section 5 (Discussion)** and shown in **Figure 5** of the original paper, E²LoRA is highly robust to these two inputs:
>
> - **Effect of Rank (Figure 5 Left):** E²LoRA consistently outperforms baselines across a wide range of rank budgets (from $r=8$ to $r=64$). The relative advantage is stable.
> - **Effect of Step (Figure 5 Right):** As detailed in our response to xxxx, the method stabilizes rapidly. Whether we use 8-512 steps, the entropy profile and resulting performance keeps relative stable.
>
> 3. **New Experiment: Robustness of Thresholds & RMI (Perturbation Analysis)**
>
> To directly address your concern about the "sensitivity to implicit assumptions in thresholding," we conducted a **noise perturbation experiment on image classification datasets**.
>
> Since the thresholds ($\tau$) derive directly from the RMI matrix, adding noise to the RMI matrix simulates errors or fluctuations in the thresholding logic. We added Gaussian noise $$\epsilon \sim \mathcal{N}(0, \sigma^2_{noise})$$ to the computed RMI matrix before partitioning and measured the impact on performance.
>
> **Results:**
>
> | noise | Params | Acc |
> | --- | --- | --- |
> | 0.001(weak) | 0.67M | 89.81 |
> | 0.005(weak) | 0.65M | 89.87 |
> | 0.01(medium) | 0.63M | 89.85 |
> | 0.05(large) | 0.55M | 89.30 |
>
> **Robustness (Small Noise):** Under small to moderate perturbations ($\sigma=0.001 \sim 0.005$), the performance fluctuates negligibly. This proves that the method is robust; it does not require "perfect" thresholds to work effectively.
>
> **Optimality (Large Noise):** When significantly larger noise ($\sigma=0.05$) disrupts the RMI structure, performance degrades. This confirms that our computed RMI matrix and the derived thresholds indeed capture a **near-optimal partitioning** structure, and randomizing this structure harms the model.

---

### Official Review · Reviewer_wUSM · 2025-10-30

**Soundness:** 3
**Presentation:** 3
**Contribution:** 3
**Rating:** 6
**Confidence:** 4

**Summary:**

This paper proposes E2LoRA, a dual-adaptive Low-Rank Adaptation framework that improves both efficiency and effectiveness via entropy-guided adaptive sharing and rank allocation.

Using a gradient-based proxy entropy analysis, the authors identify two properties in pretrained models: Local Similarity and Layer-wise Information Heterogeneity. The authors designed mechanisms that dynamically group similar layers and assign ranks proportionally to each interval’s information entropy.

**Strengths:**

1. Introduces a novel entropy-based view of LoRA sharing and rank allocation.
2.  Well-motivated and empirically supported through cross-modal experiments.
3. Simple, modular design compatible with existing LoRA and ShareLoRA.

**Weaknesses:**

1. The theoretical justification of the proposed proxy entropy metric remains largely heuristic and lacks a rigorous connection to established information-theoretic principles.
2. The empirical analysis across different model architectures and scales is limited, leaving the generality of the proposed insights insufficiently validated.
3. Certain methodological details, especially regarding entropy computation and notation, require clearer explanations to improve reproducibility.
4. The performance improvements over strong efficiency-oriented baselines such as LoRA-FA and VeRA are relatively modest in the RoBERTa base and CLIP ViT experiments, which weakens the practical significance of the gains.

**Questions:**

1. Will all models share those properties mentioned in the introduction (e.g., Larger Models (>13B), MoE Models, different series of models(Mistral, QWen, etc))?
2. Can the proposed adaptive rank allocation be combined with other LoRA variants, such as AdaLoRA, DoRA, and HiRA, and would they be complementary?
3. Could the proxy entropy approach be extended to other PEFT frameworks beyond LoRA, such as prefix-tuning or adapters?

---

> ### Author Response · Authors · 2025-11-26
> **Response to Reviewer wUSM (1)**
>
> ```
> W1: The theoretical justification of the proposed proxy entropy metric remains largely heuristic and lacks a rigorous connection to established information-theoretic principles.
> ```
>
> **Response to W1:**
>
> We thank the reviewer for pointing out the need for a clearer theoretical discussion. Below, we make explicit (i) what the proposed proxy entropy actually measures, and (ii) how it connects, in a mathematically rigorous way, to standard information-theoretic quantities.
> In summary, the proposed proxy entropy is not a heuristic approximation to Shannon entropy, but a mathematically well-defined quantity that is exactly a monotone transform of the per-parameter Frobenius norm of the gradient. Since the gradient Frobenius norm is a standard Monte Carlo estimator of the per-layer Fisher-information trace, a classical information-theoretic measure of parameter sensitivity, our proxy entropy inherits a direct and rigorous connection to established information-theoretic principles. The following paragraphs provide the detailed derivation supporting this conclusion.
>
> **(1) From proxy entropy to Frobenius norm.**
>
> For layer $\ell$, let $g\_\ell$ in $\mathbb{R}^{d\_\ell}$ be the flattened gradient and $\mu\_\ell$ its mean. The sample variance is
> $$
> \sigma\_\ell^2 = \frac{1}{d\_\ell}\sum\_{i=1}^{d\_\ell}(g\_{\ell,i}-\mu\_\ell)^2.
> $$
> Let $\tilde{G}\_\ell$ be the centered gradient tensor. Then we have
> $$
> ||\tilde{G}\_\ell||\_F^2 = \sum\_{i=1}^{d\_\ell}(g\_{\ell,i}-\mu\_\ell)^2 = d\_\ell \sigma\_\ell^2,
> $$
> then
> $$
> \log \sigma\_\ell = \log||\tilde{G}\_\ell||\_F - \tfrac{1}{2}\log d\_\ell.
> $$
> Our proxy entropy is
> $$
> H(G\_\ell) = \log(\sigma\_\ell) + \tfrac{1}{2}\log(2\pi) + \tfrac{1}{2},
> $$
> so $H(G\_\ell)$ is exactly a monotone transform of the per-parameter Frobenius norm of the centered gradient. Thus, in purely mathematical terms, our metric is not an ad-hoc heuristic but a normalized, log-compressed gradient Frobenius norm.
>
> **(2) Frobenius norm and Fisher information.**
>
> In standard settings where the gradient corresponds to the score function of a loss or log-likelihood, the Fisher information matrix per-layer is
> $$
> F\_\ell = \mathbb{E}\big[g\_\ell g\_\ell^\top\big],
> $$
> and its trace is
> $$
> \mathrm{tr}(F\_\ell) = \mathbb{E}\big[||g\_\ell||\_2^2\big].
> $$
> Hence, the squared Frobenius norm of the gradient is an unbiased empirical estimator of $\mathrm{tr}(F\_\ell)$, i.e. of the total Fisher information at layer $\ell$. Our proxy entropy $H(G\_\ell)$, being a monotone transform of $||\tilde{G}\_\ell||\_F$, is therefore also a monotone transform of an empirical estimate of the per-layer Fisher trace.
> Fisher information measures how sensitive the model is to small changes of the parameters: layers with larger Fisher information are more "informative" or important for the task. Since our proxy entropy $H(G\_\ell)$ is a monotone transform of the per-parameter Frobenius norm of the gradient, it can be viewed as a scalar summary of how much Fisher information each layer carries, which ties it directly to a classical information-theoretic notion in a simple and practical way.

---

> ### Author Response · Authors · 2025-11-26
> **Response to Reviewer wUSM (2)**
>
> ```
> W2: The empirical analysis across different model architectures and scales is limited, leaving the generality of the proposed insights insufficiently validated. And
> Q1: Will all models share those properties mentioned in the introduction (e.g., Larger Models (>13B), MoE Models, different series of models(Mistral, QWen, etc))?
> ```
>
> **Response to W2&Q1:**
>
> We appreciate the reviewer's comment regarding the generality of our insights. We fully agree that validating the proposed method across diverse architectures and scales is crucial.
>
> We respectfully highlight that our original submission already spans a broad range of architectures and modalities, supporting the universality of our insights:
>
> - **Encoder-only (NLU):** RoBERTa-Base (evaluated on GLUE).
> - **Decoder-only (NLG):** LLaMA-3.1-8B-Base & LLaMA-2-13B (evaluated on Math/Code).
> - **Vision Transformer (CV):** CLIP-ViT (evaluated on 7 image classification datasets).
> As shown in **Figure 1** and **Figure 6** of the original paper, the core patterns—*layer-wise information heterogeneity* and *local gradient similarity*—are consistently observed across these distinct architectures.
>
> To further address your concern and rigorously validate the generality of **E2LORA**, we have conducted extensive new experiments during the rebuttal phase. Specifically, we applied E2LORA to the following pre-trained weights on Math benchmarks: **Qwen3-8B** and **GPT-OSS-20B (MoE-based)**, which stands for **Different Scales, Different Families and Different Structure.**
>
> - **Universal Insights:** We have added these new visualizations in Appendix, which plot the layer-wise gradient similarity and entropy for Qwen3-8B and GPT-OSS-20B. These plots align perfectly with our original findings in Figure 1, confirming that the *local similarity* and *entropy-based heterogeneity* are intrinsic properties of LLMs, regardless of scale or architecture.
> - **Consistent Performance Gains:** The performance results (summarized in the table below and added to **Appendix**) demonstrate that E²LoRA consistently delivers on its promise of superior parameter efficiency without sacrificing performance.
>
> | Model | Method | Params | Accuracy (%) |
> | --- | --- | --- | --- |
> | Qwen3-8B (Dense) | LoRA | 7.67M | 82.49 |
> |  | E2LORA | 4.06M | 82.49 |
> | GPT-OSS-20B (MoE) | LoRA | 3.98M | 82.03 |
> |  | E2LORA | 1.75M | 83.17 |
>
> Specifically:
>
> - On **Qwen3-8B**, E²LoRA achieves the exact same accuracy (82.49%) as vanilla LoRA while using only **53%** of the trainable parameters (4.06M vs. 7.67M).
> - Even more impressively, on the large-scale **GPT-OSS-20B (MoE)** model, E²LoRA not only reduces parameter count by **56%** (1.75M vs. 3.98M) but also **surpasses** the baseline LoRA's accuracy by over 1.1 points (83.17% vs. 82.03%).
>
> These results strongly validate that E²LoRA's dual-adaptive framework is a generalizable approach. It effectively identifies and exploits parameter redundancy across different model scales, families, and even complex structures like MoE, consistently yielding a superior performance-efficiency trade-off.

---

> ### Author Response · Authors · 2025-11-26
> **Response to Reviewer wUSM (3)**
>
> ```
> W3: Certain methodological details, especially regarding entropy computation and notation, require clearer explanations to improve reproducibility.
> ```
>
> **Response to W3:**
>
> We sincerely thank the reviewer for the valuable feedback regarding the clarity of our methodological details.
>
> Below, we provide the necessary clarifications regarding the entropy computation and notation, which we have also incorporated into the revised manuscript to ensure no ambiguity remains.
>
> 1. Clarification on Proxy Entropy Computation ($H(G_l)$)
>
> In Section 3.1, we introduce "proxy entropy" to quantify the information volume of a layer's gradients. To be precise, **we treat the elements of the gradient as samples from a distribution**. Assuming the gradients follow a localized Gaussian distribution (a common assumption in weight/gradient analysis), the calculation is performed as follows (Eq. 1 in the paper):
>
> $$H(G_{l}) = \log(\sigma_{G_{l}}) + \frac{1}{2}\log(2\pi) + \frac{1}{2}$$
>
> - **Input:** $G_l$ represents the gradient of layer $l$, obtained from a mini-subset of training data (as detailed in Appendix D).
> - **Flattening:** Before computation, the tensor $G_l$ is **flattened** into a 1D vector.
> - **Standard Deviation:** $\sigma_{G_{l}}$ is the standard deviation of these flattened gradient elements.
> 2. Clarification on Notation
>
> We have standardised the notation used throughout the paper:
>
> - $N$: The total number of layers in the pre-trained model.
> - $G_l$: The gradient tensor for the $l$-th layer.
> - $RMI(G_i, G_j)$: Relative Mutual Information between layer $i$ and $j$, normalized to $[0,1]$ to measure similarity.
> - $\mathcal{S} = \{[s_k, e_k]\}$: The set of adaptive sharing intervals, where $s_k$ is the start layer index and $e_k$ is the end layer index of the $k$-th interval.
> - $H_{interval_k}$: The representative entropy for an interval, defined as the **maximum** proxy entropy among all layers within that interval ($\max_{l \in [s_k, e_k]} H(G_l)$) (5).
> - $F_k$: The normalised allocation factor for interval $k$, used to distribute the rank budget.
> - $r_{vanilla}$: The base rank used in standard LoRA (e.g., 8), used to calculate the total rank budget.
> 3. Reproducibility Assurance*: To further assist with reproducibility:
> - **Algorithmic Detail:** We refer to **Algorithm 1** (Overall Framework), **Algorithm 2** (LSS), and **Algorithm 3** (HRA) in the Appendix, which explicitly details the loop structures and variable updates.
> - **Code Release:** An anonymous GitHub repository containing our implementation is provided to allow for direct verification of our results. https://anonymous.4open.science/r/E2LoRA-23FE/e2lora.py
>
> We hope these explanations clarify the methodological details. We have updated the relevant sections in the paper to reflect these precise definitions.

---

> ### Author Response · Authors · 2025-11-26
> **Response to Reviewer wUSM (4)**
>
> ```
> W4: The performance improvements over strong efficiency-oriented baselines such as LoRA-FA and VeRA are relatively modest in the RoBERTa base and CLIP ViT experiments, which weakens the practical significance of the gains.
> ```
>
> **Response to W4:**
>
> We appreciate the reviewer's critical examination of the performance margins. While the gains on specific benchmarks like RoBERTa might appear modest due to performance saturation, we believe the practical significance of E2LORA is substantial when considering the broader landscape of **task generalization** and **methodological orthogonality**.
>
> 1. **Performance Trade-offs vs. Baselines (LoRA-FA & VeRA)**
> We would respectfully argue that the improvements are not negligible. A closer look at the comprehensive results (Table 2 & 3 in the paper) reveals distinct advantages over the mentioned baselines:
> - **Comparison with LoRA-FA:** LoRA-FA enforces a rigid freeze of the $A$ matrix across all layers, which limits expressiveness on tasks requiring nuanced adaptations. While it holds up on simpler GLUE (RoBERTa), it underperforms on more complex **NLG tasks** (e.g., GSM8K: E²LoRA +1.9%; HumanEval: +2.2%) and **Vision tasks** (+2.8% on CV datasets) with comparable parameters. This highlights E²LoRA's adaptability to varying task complexities—leveraging data-driven rank allocation to preserve plasticity, versus LoRA-FA's one-size-fits-all freeze, which struggles with generalization to intricate reasoning or multimodal scenarios.
> - **Comparison with VeRA:** VeRA's extreme parameter reduction via random frozen matrix sharing often incurs **significant performance degradation** relative to vanilla LoRA or full fine-tuning. On RoBERTa (GLUE), E²LoRA outperforms VeRA by **1.7 points** (85.57 vs. 83.83); on CLIP ViT (CV tasks), the gap is **0.9 points** (89.13 vs. 88.97), all while using ~50% fewer parameters. Unlike VeRA's static, global sharing, which applies uniform compression regardless of task demands, E²LoRA's adaptive intervals and entropy-based ranks ensure effective handling of diverse complexities, from saturated NLU to high-variance vision tasks, yielding consistent lossless compression.
> 2. **Orthogonality: E2LORA as a Universal "Plug-and-Play" Strategy**
> - Crucially, E2LORA is not merely a competing architecture but a **compression framework** derived solely from pre-trained weights (via proxy entropy and similarity). This makes it **orthogonal** to specific adapter structures.We have already demonstrated this compatibility with **ShareLoRA** (resulting in E2ShareLoRA) in the main paper.
> - **New Experiment (E2LORA + VeRA):** To directly address your comment, we combined our entropy-guided strategy with VeRA (applying adaptive interval sharing and rank to VeRA's learnable vectors). As shown in the table below, applying our E2 strategy to VeRA yields consistent improvements, proving that our insights can enhance even the most efficiency-oriented baselines.
>
>
>     | Model | Dataset | Method | Params | Accuracy/Score |
>     | --- | --- | --- | --- | --- |
>     | Roberta | GLUE | VeRA | 0.04M | 83.83 |
>     |  |  | E2VeRA | 0.013M | 84.31 |
>     | Clip-ViT | CV tasks | VeRA | 0.10M | 88.97 |
>     |  |  | E2VeRA | 0.051M | 89.13 |

---

> ### Author Response · Authors · 2025-11-26
> **Response to Reviewer wUSM (5)**
>
> ```
> Q2: Can the proposed adaptive rank allocation be combined with other LoRA variants, such as AdaLoRA, DoRA, and HiRA, and would they be complementary?
> ```
>
> **Response to Q2:**
>
> Yes, we believe the proposed adaptive rank allocation (HRA) and local similarity-based sharing (LSS) are **inherently orthogonal** to specific LoRA variants, and can be seamlessly combined with them. Our method analyzes only **the pre-trained model’s intrinsic properties,** specifically: layer-wise information heterogeneity, and inter-layer similarity. These two quantities are computed **before** inserting any LoRA module, and depend **solely on frozen pre-trained weights**. Therefore, **our interval partitioning and adaptive rank allocation do not assume any particular LoRA design** (e.g., whether $A$ is frozen like LoRA-FA, decomposed like DoRA, dynamically adjusted like AdaLoRA, or restructured like HiRA).
>
> This is reflected in our main paper, where we applied E2LORA to both **vanilla LoRA** and **ShareLoRA.** In both cases, our adaptive framework improved the parameter-efficiency trade-off, demonstrating its ability to generalize across different fundamental LoRA variants.
>
> To further validate this orthogonality with other LoRA variants, we combined our strategy with **DoRA**, creating **E²DoRA**. In this setup, we use our LSS for interval grouping and replace DoRA's uniform rank with our HRA. The results, shown in the table below, provide strong evidence for this complementarity across diverse models and tasks:
>
> | Model | Dataset | Method | Params | Accuracy/Score |
> | --- | --- | --- | --- | --- |
> | Roberta | GLUE | DoRA | 0.31M | 84.77 |
> |  |  | E2DoRA | 0.15M | 84.79 |
> | Clip-ViT | CV tasks | DoRA | 1.39M | 89.14 |
> |  |  | E2DoRA | 0.75M | 89.82 |
> | llama3.1-8B | GSM8K | DoRA | 7.14M | 69.17 |
> |  |  | E2DoRA | 2.47M | 71.03 |
>
> As the results clearly show, E²DoRA consistently matches or surpasses the performance of the original DoRA while dramatically reducing the number of trainable parameters:
>
> - On **Roberta**, E²DoRA achieves the same performance with only **48%** of DoRA's parameters.
> - On **Clip-ViT**, it improves performance while using just **54%** of the parameters.
> - Most strikingly, on **Llama3.1-8B**, E²DoRA significantly boosts the GSM8K score by nearly **2 points** (71.03 vs. 69.17), while requiring only **35%** of the original parameters.
>
> These experiments strongly suggest that E²LoRA framework as a powerful, plug-and-play **strategy**. It can be applied on top of other LoRA variants to enhance their parameter efficiency, transforming them into more cost-effective yet equally (or more) powerful methods.
>
> ---
>
> ```
> Q3: Could the proxy entropy approach be extended to other PEFT frameworks beyond LoRA, such as prefix-tuning or adapters?
> ```
>
> **Response to Q3:**
>
> Thank you for this insightful question. We believe the proxy entropy approach in E²LoRA is highly extensible to other PEFT frameworks beyond LoRA, such as prefix-tuning and adapters. As highlighted in our response to Q2, our method relies solely on analyzing the **pre-trained model’s intrinsic properties**—layer-wise information heterogeneity and inter-layer similarity—computed from frozen weights and downstream gradients. These analyses are **entirely orthogonal** to the specific adaptation mechanism like LoRA, prefix-tuning, or adapters. Since these frameworks do not alter the core pre-trained weights, our LSS and HRA can directly inform adaptive sharing and resource allocation within their structures.
>
> We are currently conducting experiments integrating E²LoRA with adapters. And we plan to include a brief discussion of this potential extension in the revised version of the paper.

---

> ### Comment · Reviewer_wUSM · 2025-11-26
>
> Thanks for your detailed rebuttal. After careful consideration of your comments, I have decided to keep my current score.

---

### Official Review · Reviewer_VLBt · 2025-10-31

**Soundness:** 3
**Presentation:** 3
**Contribution:** 2
**Rating:** 6
**Confidence:** 5

**Summary:**

This paper proposes a novel PEFT method $E^2LoRA$ to decide parameter-sharing groups across layers and allocates LoRA ranks according to a proxy-entropy measure. The approach first estimates layer-wise information and relative mutual information on a small batch of data, then proportionally distributes a global rank budget based on the estimated importance of each group. Experiments across multiple tasks demonstrate competitive or improved performance with reduced trainable parameters.

**Strengths:**

- This paper introduces a simple yet effective framework to jointly decide rank allocation and parameter sharing. The method is plug-and-play and can be applied to both vanilla LoRA and other LoRA variants, showing practical usefulness.
- This paper provides extensive experiments across multiple modalities and model scales, together with ablations that support the reported stability and generalization.

**Weaknesses:**

The comparison baselines omit several recent rank-allocation and sharing approaches (e.g., [1-5]), and the baselines used across tasks are not fully consistent, some methods appear only in selected experiments. This makes it a little difficult to characterize the contribution and relative advantage of the $E^2LoRA$.

> [1] ALoRA: Allocating Low-Rank Adaptation for Fine-tuning Large Language Models
>
> [2] DyLoRA: Parameter-Efficient Tuning of Pretrained Models using Dynamic Search-Free Low Rank Adaptation
>
> [3] IncreLoRA: Incremental Parameter Allocation Method for Parameter-Efficient Fine-tuning
>
> [4] RA-LoRA: Rank-Adaptive Parameter-Efficient Fine-Tuning for Accurate 2-bit Quantized Large Language Models
>
> [5] HydraLoRA: An Asymmetric LoRA Architecture for Efficient Fine-Tuning

**Questions:**

- What is the exact mini-subset size used to compute gradients for proxy-entropy estimation? Reporting this and adding a small ablation would strengthen the method's practicality.
- In Line 228, the interval grouping uses a greedy approach. Computing the optimal intervals should be computationally cheap; have you compared greedy vs. optimal?
- Is there any theoretical justification for using standard deviation as proxy entropy? A simple counterexample exists where a low-rank gradient matrix has higher variance than a higher-rank one, yet this metric may suggest the opposite.
- In Table 2, the parameter counts for $E^2LoRA$ on GSM8K and HumanEval appear identical. Please verify. Similarly, Table 4 and Table 5 report seemingly inconsistent parameter numbers (3.66M, 3.61M).

---

> ### Author Response · Authors · 2025-11-26
> **Response to Reviewer VLBt (1)**
>
> ```
> W1: The comparison baselines omit several recent rank-allocation and sharing approaches (e.g., [1-5]), and the baselines used across tasks are not fully consistent, some methods appear only in selected experiments. This makes it a little difficult to characterize the contribution and relative advantage of the  E²LoRA.
> ```
>
> **Response to W1:**
>
> **1. On the contribution and relative advantage of E²LoRA**
>
> We thank the reviewer for this important comment and have revised the manuscript to more clearly articulate the contribution and advantages of E²LoRA.
>
> E²LoRA introduces a **dual-adaptive sharing framework** that jointly optimizes parameter efficiency and effectiveness, built on intrinsic model properties: Local Similarity and Layer-wise Information Heterogeneity. Its core advantages are:
>
> - **Parameter efficiency as a first-class objective.**
>
>     Unlike methods focused solely on performance, E²LoRA explicitly targets efficiency. By grouping similar layers via Local Similarity-based Sharing (LSS), it consistently matches or surpasses vanilla LoRA while using roughly **50% fewer parameters** across diverse tasks. (Tables 1, 2, 3, and 8).
>
> - **Better performance–efficiency trade-off than existing sharing methods.**
>
>     Prior sharing approaches (e.g., ShareLoRA, VeRA) reduce parameters but often degrade performance due to coarse, global sharing. E²LoRA’s adaptive approach achieves comparable performance with only **~50–60%** of LoRA’s parameters. (Tables 1–3).
>
> - **Principled, data-driven sharing instead of fixed heuristics.**
>
>     Instead of fixed, manually designed sharing patterns, E²LoRA derives both sharing intervals (LSS) and ranks (HRA) directly from downstream task gradients, enabling task- and model-adaptive sharing applicable across LoRA variants.
>
>
> In summary, E²LoRA’s main novelty lies in its ability to **simultaneously** reduce the number of loras through adaptive sharing and **re-distribute** the remaining capacity via information-guided rank allocation, yielding a great efficiency–effectiveness trade-off.
>
> **2. Discussion and comparison with recent rank-allocation and sharing baselines**
>
> We appreciate the reviewer for pointing out several relevant recent works [1–5]. We add these discussion in the revised version, here we summarize the key distinctions.
>
> Overall, these methods mainly focus on **how to allocate rank within each layer** to improve performance, while E²LoRA focuses on **both** efficiency and effectiveness through **inter-layer parameter sharing and rank allocation**.
>
> - **ALoRA [1], IncreLoRA [3], RA-LoRA [4].**
>
>     These are sophisticated rank-allocation techniques. ALoRA and IncreLoRA dynamically prune, re-allocate, or increment ranks based on importance scores; RA-LoRA targets quantized models, assigning ranks to compensate quantization errors.
>
> - **DyLoRA [2].**
>
>     DyLoRA trains adapters that can operate at multiple ranks at inference time, aiming for flexible rank selection without retraining.
>
> - **HydraLoRA [5]**.
> HydraLoRA designs a specific asymmetric LoRA architecture, using a globally shared A and multiple B matrices routed by an MoE to mitigate task interference on heterogeneous data.
>
> The key distinction is that methods like [1-4] focus on intra-layer rank allocation to improve performance or flexibility, operating on individual LoRAs without sharing across layers. [5] uses a specific within-layer MoE architecture. In contrast, E²LoRA’s core novelty is **inter-layer parameter sharing** combined with adaptive rank allocation to explicitly maximize parameter efficiency.
>
> **3. On the consistency of baselines across tasks**
>
> To provide a clearer comparison, we now include **DoRA, LoRA+, AdaLoRA** in NLU and AdaLoRA in CV **to ensure major baselines (DoRA, LoRA+, LoRA-FA, VeRA) are consistently compared across tables.**
>
> As shown in the updated results, E²LoRA, which simply combines with valina LoRA, achieves highly competitive performance while utilizing ~50% trainable parameters in NLU, further validating the superior parameter efficiency and effectiveness trade-off offered by our dual-adaptive sharing framework.
>
> | NLU | params | avg. |
> | --- | --- | --- |
> | **DoRA** | 0.31M | 84.77 |
> | **LoRA+** | 0.30M | 85.62 |
> | **AdaLoRA** | 0.30M | 84.33 |
> | LoRA | 0.30M | 85.87 |
> |  E²LoRA | 0.16M | 85.57 |
>
> | CV | params | avg. |
> | --- | --- | --- |
> | AdaLoRA | 1.31M | 79.72 |
> | LoRA | 1.31M | 89.08 |
> |  E²LoRA | 0.66M | 85.57 |

---

> ### Author Response · Authors · 2025-11-26
> **Response to Reviewer VLBt (2)**
>
> ```
> Q1: What is the exact mini-subset size used to compute gradients for proxy-entropy estimation? Reporting this and adding a small ablation would strengthen the method's practicality.
> ```
>
> **Response to Q1:**
>
> Thank you for this comment regarding the practicality and data efficiency of our gradient-based initialization.
>
> 1. **Exact Mini-subset Size.**
>
>     We clarify that the exact mini-subset size used for proxy-entropy estimation is determined by the number of forward/backward steps and the training batch size. Specifically, the relationship is: Subset Size = Number of Steps * Batch Size. In our main experiments(Llama-3.1), we set the step size to 64 with a batch size of 8, resulting in a total mini-subset size of 512 samples. This represents a negligible fraction of the total training data.
>
> 2. **Ablation Study on Subset Size**
> Based on the above, the **“Effect of Step”** experiment in the Discussion section (Sec. 5) and **Figure 5 (right) is exactly an ablation of the subset size**. It demonstrates the performance variation as we increase the data used for initialization. The results show that our method is highly robust. The performance remains stable across a wide range of step values, indicating that  **E²LoRA** can effectively capture the necessary layer-wise entropy distribution with a very small amount of data. This confirms the method's high practicality and low overhead.
> 3. To address your concern and improve clarity, we have updated **Section 5 (Discussion - "Effect of Step")** to explicitly state the total number of samples used in our setup and clarified the direct relationship between "steps" and "subset size" in the discussion of Figure 5.
> ----
> ```
> Q2: In Line 228, the interval grouping uses a greedy approach. Computing the optimal intervals should be computationally cheap; have you compared greedy vs. optimal?
> ```
>
> **Response to Q2:**
>
> We thank the reviewer for the suggestion to compare the greedy interval grouping with an optimal variant. We implement an optimal interval search based on **Dynamic Programming (DP)** to compare with our greedy approach. The results confirm that the greedy strategy achieves **comparable downstream performance** to the global optimal solution. However, the greedy approach is preferred as it is **~3$\times$ faster** in search time and, crucially, avoids the need for hyperparameter tuning (e.g., penalty coefficients for interval counts), making it the more robust and efficient design choice.
>
> To rigorously validate our choice, we designed a global optimization algorithm that operates on the same Relative Mutual Information (RMI) matrix as the greedy method:
>
> - **Step 1 (Segment Cost):** For every possible contiguous interval $[i, j]$, we compute a cost defined as the negative mean similarity of layers within that interval.
> - **Step 2 (Global Objective):** We define the total objective as the sum of segment costs plus a regularization term $\lambda \cdot K$ (where $K$ is the number of intervals) to prevent over-segmentation.
> - **Step 3 (DP Search):** We employ a Dynamic Programming algorithm to strictly minimize this objective, backtracking to find the globally optimal partition $\mathcal{S}^*$ for each module type.
>
> We conduct a direct comparison on GLUE (RoBERTa) and CV (CLIP-ViT) tasks using identical training setups. The results, reporting the average performance across datasets,  are summarized below:
>
> |  |  | optimal/module | greedy(ours) |
> | --- | --- | --- | --- |
> | cv | Time | 0.6ms | 0.2ms |
> |  | Params | 0.66M | 0.66M |
> |  | Avg. Performance | 89.92 | 89.81 |
> | glue | Time | 0.6ms | 0.2ms |
> |  | Params | 0.15M | 0.16M |
> |  | Avg. Performance | 85.43 | 85.57 |
> - **Performance Parity:** The optimal grouping does not yield statistically significant improvements over the greedy approach (e.g., +0.11% on CV, but -0.14% on GLUE). This suggests that the local similarity structure of LLMs is distinct enough that a greedy strategy captures the boundaries effectively.
> - **Efficiency & Simplicity:** The DP search introduces a ~3$\times$ overhead in the grouping phase. More importantly, the DP approach requires tuning the penalty hyperparameter $\lambda$, whereas our greedy method relies on a self-adaptive threshold derived directly from RMI statistics.
>
> Given that the greedy approach offers the same performance with lower complexity and higher autonomy, we retained it as the core mechanism in E²LoRA. We have added this comparison and the DP formulation to Appendix of the revised paper.

---

> ### Author Response · Authors · 2025-11-26
> **Response to Reviewer VLBt (3)**
>
> ```
> Q3: Is there any theoretical justification for using standard deviation as proxy entropy? A simple counterexample exists where a low-rank gradient matrix has higher variance than a higher-rank one, yet this metric may suggest the opposite.
> ```
>
> **Response to Q3:**
>
> We thank the reviewer for pointing out the need for a clearer theoretical discussion. In summary, our proxy entropy is theoretically grounded because it is exactly a monotone transform of the per-parameter Frobenius norm of the gradient, which in turn is a standard and well-established quantity connected to Fisher information, sensitivity, and parameter importance in information-theoretic and optimization literature.
> It is therefore not a heuristic approximation of algebraic rank, but a principled measure of gradient energy which is conceptually different from rank and consistent with established information-theoretic principles. The detailed justification is provided below.
> We would like to clarify that our proxy entropy is not intended to estimate the algebraic rank of gradient matrices. Its purpose is to supply a single scalar per layer that reflects the overall gradient energy relevant to downstream adaptation, a quantity conceptually distinct from rank.
>
> **(1) What proxy entropy measures.**
>
> For layer $\ell$, let $g\_\ell$ in $\mathbb{R}^{d\_\ell}$ be the flattened gradient and $\mu\_\ell$ its mean. The sample variance is
> $$
> \sigma\_\ell^2 = \frac{1}{d\_\ell}\sum\_{i=1}^{d\_\ell}(g\_{\ell,i}-\mu\_\ell)^2.
> $$
>
> Let $\tilde{G}\_\ell$ be the centered gradient tensor. Then,
> $$
> ||\tilde{G}\_\ell||\_F^2 = \sum\_i \^{d\_\ell} (g\_{\ell,i}-\mu\_\ell)^2 = d\_\ell \sigma\_\ell^2,
> $$
> then we have
> $$
> \log \sigma\_\ell = \log||\tilde{G}\_\ell||\_F - \tfrac{1}{2}\log d\_\ell.
> $$
>
> Our proxy entropy is defined as
> $$
> H(G\_\ell) = \log(\sigma\_\ell) + \frac{1}{2} \log(2\pi) + \frac{1}{2},
> $$
>
> so $H(G\_\ell)$ is exactly a monotone transform of the per-parameter Frobenius norm of the (centered) gradient:
> $$
> H(G\_\ell) = \log ||\tilde{G}\_\ell||\_F - \frac{1}{2} \log d\_\ell + \text{const.}
> $$
> Therefore, what we are really measuring is: The total gradient energy $||\tilde{G}\_\ell||\_F^2$, normalized by layer size (via division by $d\_\ell$), and then log-compressed to stabilize scale.
> This is closely aligned with standard practice in the literature, where Frobenius norms or related second-order measures (e.g., the trace of the Fisher information, which equals the expected squared gradient norm) are used as indicators of parameter / layer importance, sensitivity, or curvature.
>
> **(2) Why is rank irrelevant here.**
>
> We never assume nor require any monotonic relation between rank and variance/Frobenius norm. Indeed, for
> $$
> A=\alpha uv^\top \ (\text{rank}=1),\quad B=\varepsilon I\_d \ (\text{rank}=d),
> $$
> we have
> $$
> ||A||\_F^2=\alpha^2,\quad ||B||\_F^2=d\varepsilon^2,
> $$
> and choosing $\alpha\gg\sqrt{d} \varepsilon$ gives $\text{rank}(A)<\text{rank}(B)$ but $||A||\_F>||B||\_F$. Thus variance/Frobenius norm cannot be interpreted as a rank estimator, which is precisely why our method refrains from doing so.
> Importantly, our goal in adaptive rank allocation is to assign more LoRA capacity to more important layers, i.e., layers with larger proxy entropy indicating higher gradient energy, rather than to layers whose gradients have higher algebraic rank.
>
> **(3) Effective rank in practice.**
>
> Real neural-network gradients are typically numerically full-rank; discussions of “rank” usually refer to effective rank, which depends on singular-value thresholds. Our design intentionally focuses on a robust, threshold-free energy measure instead of algebraic or effective rank.
>
> **(4) Proxy nature and empirical support.**
>
> We explicitly refer to $H(G\_\ell)$ as a proxy entropy: an entropy-inspired, scale-sensitive second-order statistic. Across all evaluated modalities (NLU, NLG, vision) and model families, this proxy produces stable layer orderings and consistently superior parameter–performance trade-offs, validating its practical suitability for adaptive sharing and rank allocation.
>
> ---
> ```
> Q4: In Table 2, the parameter counts for  on GSM8K and HumanEval appear identical. Please verify. Similarly, Table 4 and Table 5 report seemingly inconsistent parameter numbers (3.66M, 3.61M).
> ```
>
> **Response to Q4:**
>
> We thank the reviewer for pointing out the inconsistency in the reported parameter counts.
> After reviewing the logs, we have confirmed that the parameter count for GSM8K is 3.61M, while for HumanEval it is 3.66M. We correct these values throughout the manuscript to show the correct numbers.

---

### Author Response · Authors · 2025-12-03
**Summary of Rebuttal**

Dear Area Chairs, Senior Area Chairs, and Program Chairs,

We sincerely thank all reviewers, ACs, and PCs for your time and effort in the review process. Below, we summarize our core contributions and the key outcomes of the rebuttal.

1. **Core Contributions**

This paper proposes **E²LoRA**, a novel **adaptive** framework that **automatically determines inter-layer parameter sharing partitions and allocates ranks within shared intervals**, thereby maximizing parameter efficiency while **maintaining or even surpassing** the performance of standard methods.

2. **Summary of Strengths**

We sincerely appreciate the reviewers' recognition of our work's novelty and practical value, as highlighted below:

- **Novel entropy-based perspective.** "Introduces a novel entropy-based view... well-motivated and empirically supported" (Reviewer wUSM); "This perspective is somewhat original" (Reviewer LoWT); "A novel PEFT method" (Reviewer VLBt).
- **Simple yet effective framework.** "Simple yet effective framework to jointly decide rank allocation and parameter sharing" (Reviewer VLBt); "Simple, modular design" (Reviewer wUSM).
- **Strong cross-modal validation.** "Provides extensive experiments across multiple modalities and model scales" (Reviewer VLBt); "Well-motivated and empirically supported through cross-modal experiments" (Reviewer wUSM).
- **Plug-and-play compatibility.** "The method is plug-and-play" (Reviewer VLBt); "The dual-adaptive scheme is nearly plug-and-play with existing LoRA pipelines" (Reviewer CKkS).
3. **Key Concerns Resolved**

In our rebuttal, we conducted extensive new experiments and provided detailed clarifications to address the concerns raised by the reviewers. Below, we summarize the main shared concerns and our detailed responses:

- **Generality across Architectures and Scales.** We extended our evaluation to **Qwen3-8B** and **GPT-OSS-20B (MoE)**. E²LoRA consistently demonstrates superior efficiency-performance trade-offs regardless of model family or architecture. Notably, on the MoE model (GPT-OSS-20B), E²LoRA reduces parameters by **56%** (1.75M vs 3.98M) while **improving accuracy by +1.14%** over vanilla LoRA. (Please refer to Reviewer wUSM: Weakness 2; Reviewer CKkS: Weakness 4).
- **Orthogonality and Extensibility.** We demonstrated that E²LoRA is a universal framework that can compatible with other PEFT methods. By applying our strategy to **DoRA,** **VeRA, and LoRI**, we created **E²DoRA,** **E²VeRA, and E²LoRI**. E²DoRA on GSM8K achieves a score of **71.03** vs DoRA's **69.17**, while using only **35%** of the parameters (2.47M vs 7.14M), proving our method acts as a "universal enhancer" rather than just a competitor. Furthermore, E²LoRI-S further reduces parameters by ~50% to **0.007M** while improving NLU accuracy (**+0.7** points). (Please refer to Reviewer wUSM: Weakness 4; Reviewer LoWT: Weakness 3; Reviewer CKkS: Weakness 2).
- **Comparison with Parameter-Matched Baselines.** We added comparisons against recent methods, including DoRA, LoRA+, and  AdaLoRA. In parameter-matched settings, E²LoRA consistently outperforms baselines. For instance, on NLU tasks, E²LoRA (0.16M params) surpasses the **parameter-matched** DoRA (0.17M params) with a higher average score (**85.57 vs 84.76**). (Please refer to Reviewer CKkS: Weakness 1; Reviewer LoWT: Question 3).
- **Theoretical Grounding and Robustness.** We clarified that our proxy entropy is a monotone transform of the **gradient Frobenius norm**, serving as an empirical estimator of the **Fisher Information trace**, thus firmly grounded in information theory. New ablation studies confirm the method is highly **robust** to data subset size (stable across 8-512 steps) and resilient to noise perturbations in the RMI matrix. (Please refer to Reviewer VLBt: Question 3; Reviewer wUSM: Weakness 1; Reviewer LoWT: Weakness 2 & 4).
- **Computational Overhead.** We verified that the overhead is negligible (**\~30 seconds**, <2% of training time) and is incurred only once per task, justifying the significant parameter savings (\~50%) and performance gains. (Please refer to Reviewer LoWT: Weakness 1).
- **Coverage of Related Work.** We have expanded our discussion to include recent works such as BSLoRA and IncreLoRA. We clarified the distinctions, highlighting that E²LoRA uniquely targets adaptive inter-layer parameter sharing and efficiency, whereas other methods typically focus on initialization or intra-layer rank allocation. (Please refer to Reviewer VLBt: Weakness 1; Reviewer CKkS: Weakness 3).

We believe these additional results and clarifications strengthen our submission and help clarify the issues raised, highlighting E²LoRA as a principled, robust, and highly practical contribution to the PEFT landscape.

We hope this summary assists you in quickly grasping the core discussions and the value of our work. Thank you again for your time and consideration.

Best regards,

Paper7059 Authors

---

### Meta-Review · Area_Chair_adiG · 2026-01-03

**Summary:**

The paper's initial scores are 6622. I am inclined to believe that after the good rebuttal put up by the authors, their scores would have been improved to at least a 6844. Note that this is based on my own judgment; after all, we are well-aware that some reviewers do not care to read the rebuttals. Back to the main topic, my sense is that LoWT and CKkS brought up some points that the authors addressed well. Hence, I recommend an accept, but am fine if my decision is bumped down.

**Reviewer Concerns:**

Most reviews were well responded. The only unsatisfactory part is wUSM's point on theoretical justification.

**Reviewer Scores:**

LoWT's concerns about additional complexity, robustness to proxy entropy measure, parameter-matched comparisons, and hyperparameter sensitivity were well addressed. CKks brought up performance and efficiency questions, baseline questions, and model choices. The authors responded well.

---

### Decision · Program_Chairs · 2026-01-26

Accept (Poster)